# Local mutational diversity drives intratumoral immune heterogeneity in non-small cell lung cancer

Qingzhu Jia[1,2], Wei Wu[3], Yuqi Wang[4], Peter B. Alexander[5], Chengdu Sun[1,2], Zhihua Gong[1,2], Jia-Nan Cheng[1,2,6], Huaibo Sun[4], Yanfang Guan[4], Xuefeng Xia[4,7], Ling Yang[4], Xin Yi[4], Yisong Y. Wan[8], Haidong Wang[3], Ji He[9], P. Andrew Futreal[10], Qi-Jing Li[5,6] & Bo Zhu[1,2]

Combining whole exome sequencing, transcriptome profiling, and T cell repertoire analysis, we investigate the spatial features of surgically-removed biopsies from multiple loci in tumor masses of 15 patients with non-small cell lung cancer (NSCLC). This revealed that the immune microenvironment has high spatial heterogeneity such that intratumoral regional variation is as large as inter-personal variation. While the local total mutational burden (TMB) is associated with local T-cell clonal expansion, local anti-tumor cytotoxicity does not directly correlate with neoantigen abundance. Together, these findings caution against that immunological signatures can be predicted solely from TMB or microenvironmental analysis from a single locus biopsy.

[1] Institute of Cancer, Xinqiao Hospital, Third Military Medical University, Chongqing 400037, China. [2] Chongqing Key Laboratory of Tumor Immunotherapy, Chongqing 400037, China. [3] Department of Cardiothorathic Surgery, Southwest Hospital, Third Military Medical University, Chongqing 400038, China. [4] Geneplus-Beijing Institute, Beijing 102206, China. [5] Department of Immunology, Duke University Medical Center, Durham 27710 NC, USA. [6] Biomedical Analysis Center, Third Military Medical University, Chongqing 400038, China. [7] Houston Methodist Research Institute, Houston 77030 TX, USA. [8] Department of Microbiology and Immunology, Lineberger Comprehensive Cancer Center, University of North Carolina at Chapel Hill, Chapel Hill 27514 NC, USA. [9] GeneCast Biotechnology Co., Ltd, Beijing 102206, China. [10] Department of Genomic Medicine, The University of Texas MD Anderson Cancer Center, Houston, TX, USA. These authors contributed equally: Qingzhu Jia, Wei Wu. Correspondence and requests for materials should be addressed to Q.-J.L. (email: Qi-Jing.Li@Duke.edu) or to B.Z. (email: bo.zhu@tmmu.edu.cn)

I mmune checkpoint blockade such as Nivolumab has delivered unprecedented success in treating non-small cell lung carcinoma (NSCLC) in both first-line[1,2] and second-line[3–6] treatments to extend overall survival. However, while prolonged and durable responses can be achieved, only a small percentage of patients experience such clinical benefits[7]. Pre-primed cytotoxic T-lymphocyte (CTL) infiltration and pre-existing cytolytic activity have been established as indicators of a greater likelihood of response to immunotherapy[8]. Accordingly, various biomarkers for measuring anti-tumor reactivity in fresh or archival biopsies have demonstrated a certain degree of prognostic or predictive value in many solid tumor types[4,9,10]. These biomarkers mainly focus on characterizing unique features of pre-existing anti-tumor reactivity in the tumor microenvironment (TME) and include *programmed death-ligand 1* (*PD-L1*) expression[3], CD8$^+$ T-cell infiltration[11], T-cell repertoire clonality[8], and panel-based immunological signatures[12]. Conceptually, "non-self" neoantigens generated by mutated cancer cells could trigger their own elimination by T cells, implying that a higher mutational burden should result in a stronger anti-tumor immune response. The prognostic significance of total mutational burden (TMB)[13,14], deficient mismatch repair (dMMR), and micro-satellite instability status in predicting response to immunotherapy also supports the applicability of this theory[15].

However, emerging evidence continues to challenge this association between mutational load and the anti-tumor response. Although *PD-L1* is thought to be induced by *interferon-γ* (*IFN-γ*)-mediated immune responses, TMB was not correlated with *PD-L1* expression in the CheckMate 026 NSCLC trial[14]. For patients with dMMR status, only 62% experienced a clinical benefit to anti-PD1 therapy[16]. Furthermore, Charoentong et al.[17] demonstrated that TMB plays a less-than-expected role in determining TME immunogenicity. These inconsistencies suggest that more comprehensive analyses of TME immunophenotypes are needed.

In this study, multiple systemic approaches are employed to assess immunogenicity beyond neoantigen abundance and mutational burden. Since spatially heterogeneous immunoreactivity might weaken the predictive value of current biomarkers, we also sample multiple regions from each individual NSCLC tumor. Moreover, a machine learning algorithm is developed to integrate 278 variables for depicting local anti-tumor responses. This study reveals that immunoreactivity is spatially heterogeneous between different tumor regions. It also suggests that the accurate identification of candidates for immunotherapy might be improved by more comprehensively measuring local tumor attributes, in comparison to solely focusing on TMB or neoantigen load as is the current practice.

## Results

**Mutational burdens correlate with local T-cell expansion**. To comprehensively assess the correlation between non-synonymous somatic mutations and immune responses, we recruited 15 newly diagnosed NSCLC patients who had surgery with curative intent (Supplementary Data). Multiple biopsy samples were collected from each primary tumor mass according to a previously established method[18]. For each sample, we performed multiple genomic and immunogenomic assays including whole-exome sequencing (WES), transcriptome profiling (RNA-sequencing (RNA-seq)), and T-cell repertoire sequencing. The cellular purity of tumor cells in each biopsy was calculated based on WES data (Supplementary Data). In total, whole-exome and T-cell repertoires from 57 loci of 15 patients, and transcriptome profiling from 44 loci of 12 patients, were analyzed in an integrated bioinformatics pipeline (Fig. 1a and Supplementary Fig. 1). Tumor samples and peripheral blood mononuclear cell controls were sequenced at median 115-fold coverage across exome

capture loci. In total, 2462 non-synonymous somatic mutations were identified (Supplementary Data). Numbers of genomic mutations varied substantially in different loci among patients, ranging from 2 to more than 380 per sample.

Non-synonymous somatic mutation in the coding region of a gene (Fig. 1b and Supplementary Fig. 2, upper panel) can generate presentable neoantigens (Fig. 1c and Supplementary Fig. 2, middle panel), which can be recognized by T cells with structurally divergent and antigen-specific T-cell receptors (TCRs). Upon antigen-stimulated activation, T cells expand within the tumor to create an effector pool to execute their cytolytic function and control tumor growth. Our previous studies suggested that intratumoral T-cell repertoires are enriched with tumor antigen-specific and clonally expanded T cells[19]. We therefore performed TCR repertoire sequencing of spatially distinct tumor samples and analyzed high-frequency T-cell clones along with predicted neoantigens. While a few non-synonymous mutations and neoantigens were shared among different tumor loci from the same patient, neoantigen heterogeneity was also detected for each patient (Fig. 1b, c). Accordingly, measured by the similarity of CDR3B sequences (the major structural domains for antigen recognition[20,21]), high-frequency[19] TCRs were also distinct among individual patients. Within each patient, identical dominant CDR3B clonotypes were identified across different loci. However, because unique mutations in different loci lead to neoantigen heterogeneity, a range of T-cell reactivity was also reflected by unique high-frequency T-cell clones in different tumor locations (Fig. 1d and Supplementary Fig. 2 lower panel, Supplementary Data for high-frequency CDR3 clones and for all identified clones).

It has been suggested that higher TMB favors neoantigen-specific T-cell infiltration and oligoclonal expansion[22]. Based on CDR3 TCR sequences, we calculated the Shannon entropy index (Fig. 1e, $R = 0.3448$, \*\*$p = 0.00861$, Pearson's correlation) and Simpson diversity index (Fig. 1f, $R = 0.3425$, \*\*$p = 0.00911$, Pearson's correlation) for each sampled tumor site and identified a statistically significant but moderate positive correlation between local T-cell repertoire clonality and TMB (Supplementary Data) or neoantigen frequency (Supplementary Figs. 3a, b and Supplementary Data). This suggests that, within the TME, higher TMB may drive more diversified T-cell clonotypes into proliferation.

**Mutational burden is loosely related to cytolytic activity**. It is known that changes in a local TCR repertoire correlate with intratumoral CD8$^+$ T-cell activation[23]. To determine whether T-cell neoantigen-stimulated oligoclonal expansion also leads to augmented CTL function, we first examined the local production of effector molecules released by cytolytic granules, such as T-cell granzyme-A (*GZMA*) and IFN-γ. Unlike T-cell activation as measured by clonotype dynamics, both *GZMA* (Fig. 1g; $R = 0.0954$, $p = 0.5378$, Pearson's correlation) and IFN-γ (Fig. 1h; $R = 0.1688$, $p = 0.2733$, Pearson's correlation) production showed weak association with the local TMB. A similar uncoupling between TMB and T-cell activation was reported in immunotherapy-naive melanoma[24] and pancreatic cancer[25] cohorts. Previous studies have used the geometric mean of *GZMA* and *perforin 1* (*PRF1*) as a surrogate measure of CTL activity[26]. However, the correlation between this parameter and TMB was also weak and statistically insignificant (Fig. 1i; $R = 0.1199$, $p = 0.4381$, Pearson's correlation). Furthermore, instead of TMB, when the loads of neoantigen predicted at the cutoff of human leukocyte antigen (HLA)-binding affinity at 50 nM were used, we also failed to find any significant correlation with local cytotoxicity (Supplementary Figs. 3c-e).

Recently, McGranahan et al.[22] reported that high clonal mutational burden and low subclonal mutational heterogeneity

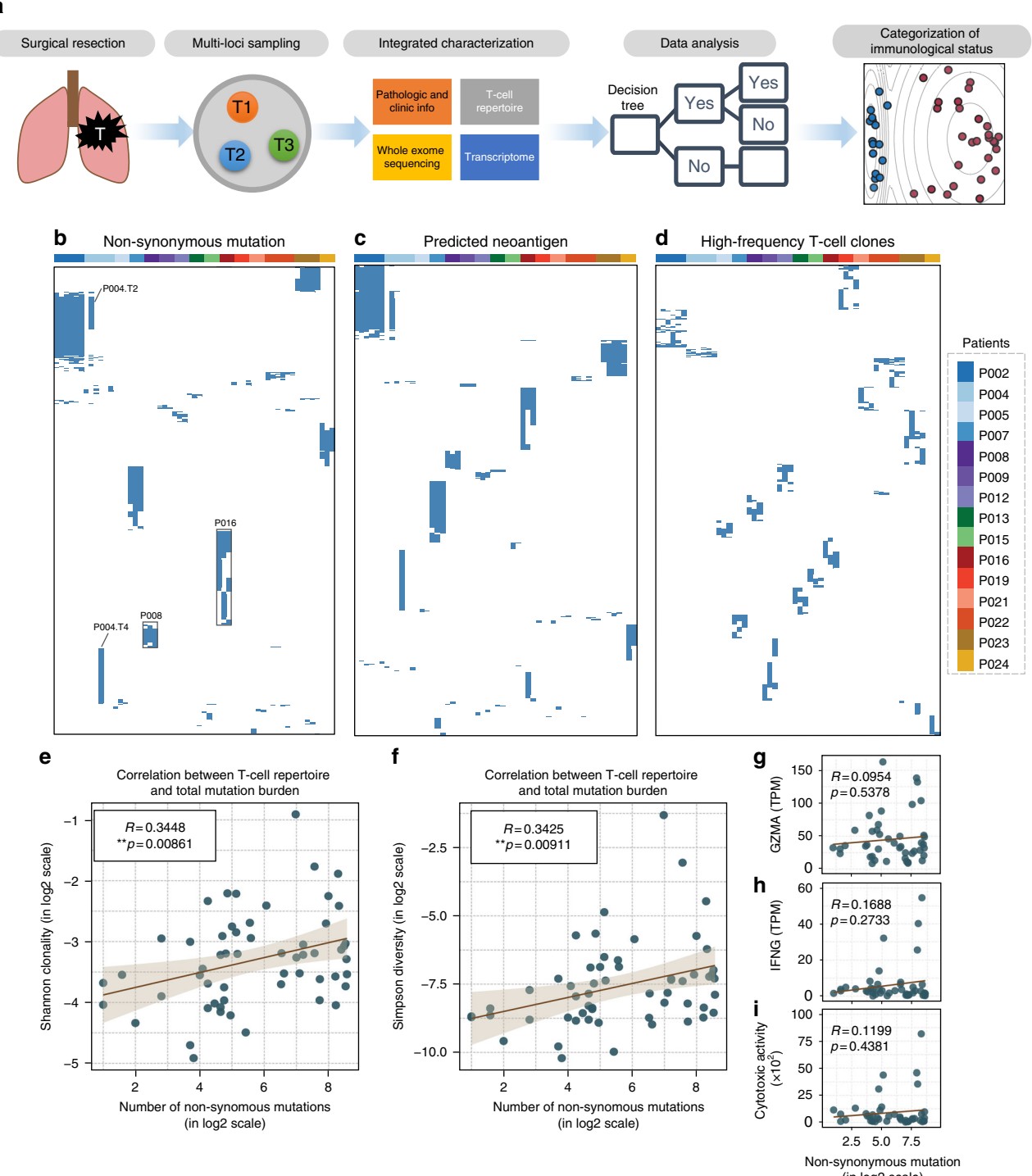

**Fig. 1** Total mutational burden correlates with T-cell clonality. **a** Schematic of sampling strategy and experimental workflow. Tissues from multiple loci within the whole tumor were resected and subjected to high-throughput sequencing. Non-synonymous mutations, HLA typing, predicted neoantigens, transcriptomic profiling, and T-cell repertoire were analyzed to characterize TME heterogeneity. **b–d** Heat maps depicting the inter-population and intratumoral distribution of non-synonymous mutations, predicted neoantigens (with binding strength <500 nM), and dominant T-cell clones (frequencies ≥0.5%) in all sequenced subjects; presence (blue) or absence (white) is indicated for every tumor focus. Samples were grouped according to individual patients. **e**, **f** Scatterplot showing correlation between total mutation load and expanded properties of the T-cell repertoire. T-cell clonality (**e**) and Simpson diversity index (**f**) were used to depict the T-cell repertoire composition. Enrichment of highly expanded clones results in higher values for clonality and Simpson diversity. R coefficient of Pearson's correlation. Shaded area, 95% confident interval for the correlation. **g–i** Correlation between mutation load (in log2 scale) with expression of interferon-gamma, granzyme-A, and cytolytic activity (measured as the geometric mean of granzyme-A with perforin 1) in log2 of transcript per kilobase million (TPM)

correlate with superior prognosis in an immunotherapy-naive NSCLC cohort. This feature can also be used to identify patients who are likely to benefit from checkpoint blockade therapies[22]. To exclude possible subclonal mutations as a confounding factor in our analysis, we dissected clonal mutational burdens using two independent methods[22,27]. Subsequent subtraction of subclonal mutations failed to improve the correlation between TMB and local immune cytotoxicity: clonal mutational burdens remained poorly associated with inflamed immune signatures (Supplementary Fig. 4). Finally, we used a more sophisticated eight gene panel designed in the POPLAR trial[4] as a biomarker to reflect effector T-cell infiltration (*CD8A*, *CXCL9*, and *CXCL10*) and IFN-γ-associated cytotoxicity (*IFNG*, *GZMA*, *GZMB*, *EOMES*, and *TBX21*). While the expression of this gene set strongly correlated with the responsiveness of NSCLC patients to anti-PD-L1 treatment[4], in our cohort, no significant association was observed between this POPLAR biomarker panel and the local TMB (Supplementary Fig. 5).

**Spatially heterogeneous tumor immune microenvironments**. To further improve accuracy in assessing the tumor immune microenvironment, we developed a machine learning approach to integrate multidimensional immune-related variables for data mining. Using a random forests algorithm, we classified the immune microenvironment of tumor loci with 278 input variables, which include neoantigen loads, T-cell repertoire clonality, expression of 31 well-categorized genes that are critical for immune regulation and antigen presentation (such as *ICOS*, *IDO1*, *PDCD1*, *TIGIT*, and *LAG3*), 217 enrichment scores outlining 217 signal transduction pathways regulating various immune functions, and the abundance of 28 subpopulations of infiltrating immune cells (such as activated CD8$^+$ T cells, myeloid-derived suppressor cells (MDSCs), and regulatory T cells (Tregs)) (Supplementary Data)[17,28]. Except neoantigen and TCR clonality assessments, all other 276 parameters were derived from data generated by RNA-seq of bulk tissues from each individual tumor loci (transcripts per million (TPM) in Supplementary Data). Specifically, quantitative measurements of signaling pathway activation and immune cell infiltration were generated by single-sample Gene Set Enrichment Analysis (ssGSEA)[29,30], which has been applied in a number of studies to infer from RNA profiling data the relative level of immune cell infiltration[17,31–33]. Using this method, we transformed the transcriptomic expression data into a normalized score to represent the activation status of specific pathways or the relative abundance of specific immune cell types (Supplementary Data). We then reduced this 278-dimension matrix to a two-dimensional plot, in which the proximity of two dots represents the immune microenvironment similarity of two sampled NSCLC tissues. Eventually, using density contours generated from Gaussian maximum fitting, we were able to visualize the immune phenotype of the spatial TME as a confined location in the contour plot which was termed the "immune map". On this immune map, 44 samples from 12 patients were categorized as immunologically "hot" versus "cold" loci, thereby separating these tumor tissues based on their cytolytic activity levels[26] (Fig. 2a).

To validate its analytic power, we tested whether the location of certain TME on the immune map could properly predict the local expression of the eight biomarkers established in the POPLAR trial[4]. Indeed, all eight genes associated with CD8$^+$ T-cell infiltration and cytotoxicity were expressed at significantly higher levels in the hot area compared to the cold area (Fig. 2b, c). Recently, Trajanoski and colleagues[17] developed a score system to predict tumor responsiveness to CTLA-4 (cytotoxic T-lymphocyte-associated protein 4) and PD1 blockade. In their analysis, 26 scoring variables, including 4 effector cell types, 2 suppressor cell types, 10 immune checkpoint molecules, and 10

genes involved in antigen presentation, were integrated for immune TME evaluation. When these 26 parameters were examined using our immune map algorithm, while activated CD4$^+$ T-cell abundance was not statistically distinct in these two areas, the remaining 25 parameters representing immunogenicity were substantially inflamed in the hot area (Supplementary Fig. 6). Taken together, we reasoned that our immune map algorithm is a suitable platform to categorize the immune TME based on its local anti-tumor cytotoxicity.

Using this immune map, we assessed whether TMB can impact local immunogenicity. Importantly, we found that the tumor immunogenicity cannot be directly predicted by either local mutation burden or neoantigen loads: tumor loci with high TMB can be found in the cold area and loci with low TMB can be categorized as immunologically hot tissues (Fig. 2d). Furthermore, immune mapping also enabled the identification of intratumoral immunological heterogeneity for each individual (Fig. 2e, detailed labeling in Supplementary Fig. 7). Of the 44 tumor loci with complete immunogenomics analysis, 28 loci were allocated into the immunologically hot area and 16 loci were designated as immunologically cold (Fig. 2f, upper panel). Notably, half of the NSCLC patients (6 of 12) harbor both immunological hot and cold areas synchronously in a single tumor (Fig. 2f, lower panel). When we assessed *PD-L1* expression, which is known to be upregulated upon IFN stimulation[34] and serves as a biomarker for patient stratification for anti-PD1 immunotherapy[3], we found that both messenger RNA (mRNA) and protein levels of *PD-L1* were heterogeneous among different tumor loci and did not correlate with local TMB (Fig. 2g and Supplementary Fig. 8). Furthermore, for these patients, intratumoral immune heterogeneity was also reflected in the differential expression of a range of immunoregulatory and effector molecules (Fig. 2h). Based on these results, we conclude that, in NSCLC tumors, anti-tumor cytotoxicity can be localized into different regions such that local mutational burden and neoantigen variety cannot effectively predict the overall state of immune activation.

**Mutational heterogeneity is related to immune heterogeneity**. The uncoupling of local tumor antigenicity (TMB) and cytotoxicity suggests that T-cell activities are subjected to additional immunosuppressive regulation. In the TME, immunosuppression can be generated through at least two possible mechanisms: (1) tumor-infiltrated T cells can be suppressed by feedback-induced checkpoint molecule expression (Fig. 2h) and immunoregulatory cell recruitment; and (2) different mutations may play distinct roles in modifying the immune response[26].

To investigate the regulation of suppressive immune cell populations, we composed a heatmap to visualize the relative abundance of 28 infiltrating immune cell populations (Fig. 3a). We observed two major features. (1) immune cell infiltration was also spatial and isolated, resulting in heterogeneous immune cell compositions between different loci within an individual tumor. This feature was validated by flow cytometry using fresh isolated biopsies sampled from multiple loci (Fig. 3b). (2) Immunologically hot tumor regions overall had a higher abundance of immune cell infiltration, including cells executing anti-tumor reactivity (e.g., activated CD8$^+$ T cells, type 1 T helper cells, and activated dendritic cells) and cells delivering pro-tumor suppression (e.g., MDSCs, regulatory T cells, immature dendritic cells, and neutrophils). Pearson's correlation analysis showed that the abundances of these two categories of cells was positively associated within a local environment (Fig. 3c). This observation suggests the presence of a feedback mechanism such that the recruitment or differentiation of cells specialized for immune suppression may be facilitated by anti-tumor inflammation.

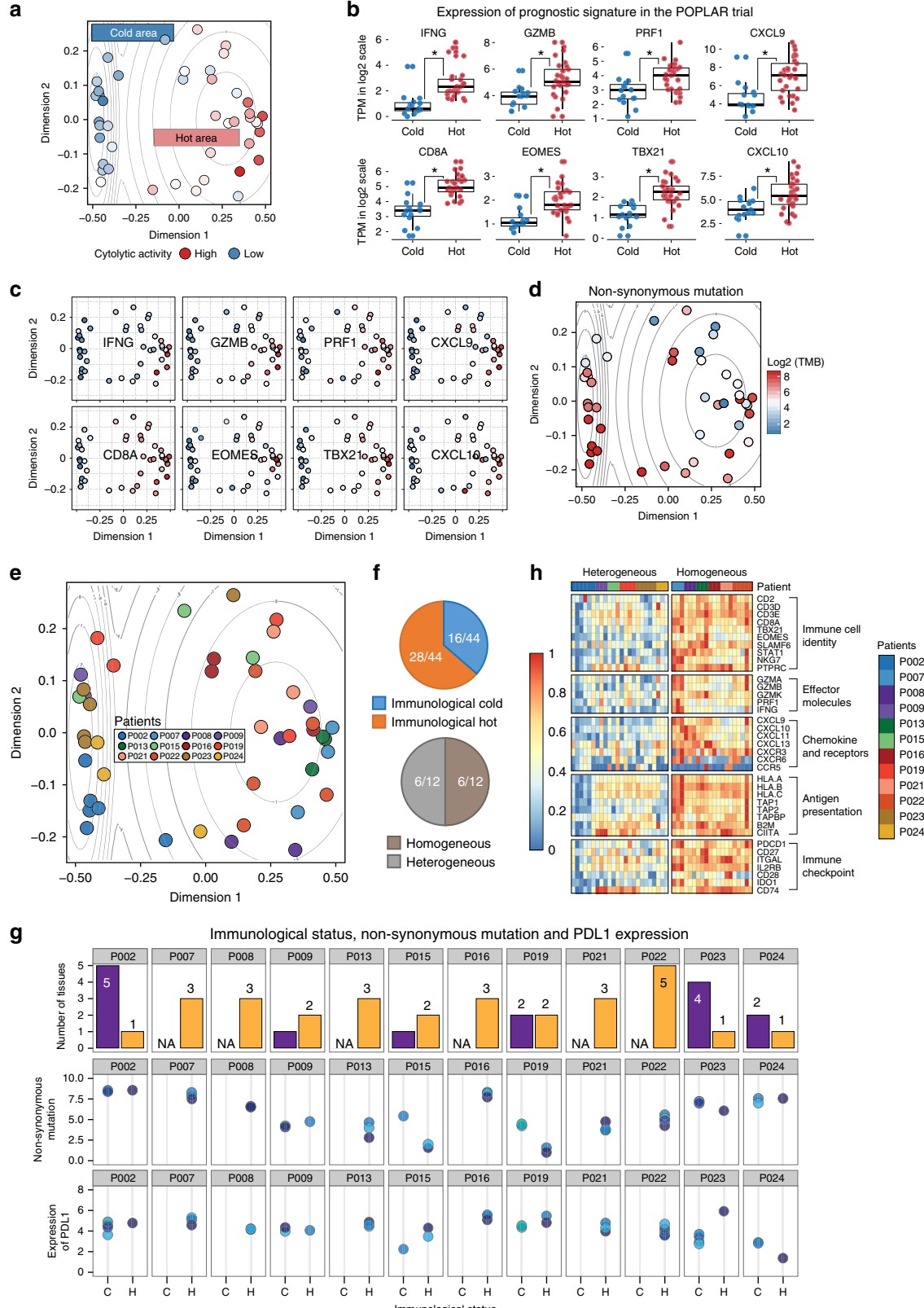

Since anti-tumor inflammation is largely determined by tumor antigen-specific T cells, and infiltrating T-cell clonality is associated with neoantigen load, we analyzed whether the heterogeneity of immune cell infiltration was related to the heterogeneity of TMB. Two different measurements, "coefficient of variance" and "intratumoral heterogeneity" (ITH)[22], were

employed to quantify mutational heterogeneity. An average correlation coefficient value for every pair of immune cell frequencies was used to quantify the divergence of immune cell infiltration. Coefficient of variance preferentially weighs the total number of mutations. Although falling short of statistical significance, genomic heterogeneity was associated with inter-

**Fig. 2** Machine learning classifies TME into hot and cold immunophenotypes. **a** Visualization based on two-dimensional coordinates from multidimensional scaling (MDS) of proximity matrix from the input variables in NSCLC. Color indicates cytolytic activity (product of *PRF1* and *GZMA*) for each sampling site. To categorize the samples by an unsupervised method, the Gaussian expectation maximization algorithm was employed to perform categorization under Gaussian mixture models. The contour shows the estimated probability density for the two categories. Left, cold area; right, hot area. **b**, **c** Expression profiling of prognostic genes in the POPLAR study for hot and cold areas. Expression values are transformed TPM format and in log2 scale. Horizontal bar in boxplot, median value. Statistics based on two-tailed Mann–Whitney *U*-test. **d** Total mutational burden in the categorization. **e** Samples from each individual are labeled by color. Contours guiding immunologic categorization are shown as in **a**. **f** Upper panel, pie chart showing the proportion of hot vs cold area samples in all sequenced tissues. Lower panel, proportion of patients harboring multiple distinct immunological statuses simultaneously. Homogeneous, patients with one kind of immunological status; Heterogeneous, patients with both immunological statuses. **g** Correlation among immunological status, TMB, and PD-L1 expression. Upper panel, the proportion of immunological statuses found in each individual. Middle and lower panels, TMB and PD-L1 expression for each sample. For each individual, the samples in the middle and lower panels were labeled in the same colors. **h** Heatmap showing the expression of feasible immunotherapy-predictive immune genes for 44 NSCLC samples. Expression values for each gene are normalized into *z*-scores

locus divergence of immune cell infiltration (Fig. 3d, left panel; $R = 0.5433$, $p = 0.06793$, Pearson's correlation). ITH levels, which weigh more on unique mutations among different sampling foci, were positively associated with the inter-locus divergence of immune cell infiltration (Fig. 3d, right panel; $R = 0.6435$, $*p = 0.0239$, Pearson's correlation).

To explore whether TME immune heterogeneity is impacted by the nature of mutations, we traced the subclonal architectures of non-synonymous somatic mutations for all 15 patients (Supplementary Fig. 9). Specifically, for patients with intratumoral immune heterogeneity, 5 out of 6 (83%, except patient 024) displayed a dichotomy of mutations that separates cold versus hot regions into divergent evolutionary directions: other than homogenous (progenitor) mutations at every locus, hot and cold tumor regions did not share any common mutations (Fig. 3e). This suggests that the functional nature of a mutation-carrying protein may also play a role in determining the immunogenicity of neoantigens. Taken together, we propose that tumor-intrinsic immunomodulation, which can be provoked by certain mutations, combined with spatial infiltrating complexity of immune cells, which may be elicited through a feedback mechanism after T-cell activation, determines the ultimate immunophenotype of a given tumor locus.

## Discussion

Here we used a combinational omics strategy to comprehensively evaluate the TME immunophenotype of NSCLC. Critically, we sampled multiple loci within a single tumor to reveal the spatial heterogeneity of the TME. While the intratumoral heterogeneity of somatic mutations has been well studied[35–38], the intratumoral heterogeneity of anti-tumor reactivity is currently under-appreciated. From a clinical perspective, just as genomic intra-tumoral heterogeneity poses a challenge to traditional targeted therapies, spatial distribution of the immune microenvironment also provides a challenge for targeted anti-cancer immunotherapy. For example, through a few successful clinical trials, a batch of biomarkers has been developed to sub-group and stratify patients[9,10,39]. However, most current biomarkers are supported by data collected from a single locus biopsy and based on the assumption that the TME is homogeneous. The current study strongly suggests that the predictive power of single locus biopsy is limited and a more comprehensive analysis of tumor immune niches by multiple sampling may be necessary. Moreover, we also found that immune suppressive machineries, such as *PD-L1*, are diversely expressed throughout the tumor. This suggests that monotherapies that target a single immunosuppression mechanism are unlikely to have the intended overall response for NSCLC patients. This idea is supported by a recent NSCLC patient analysis of the Rizvi cohort[13], in which superior outcomes

to anti-PD1 therapy were associated with more homogenous genomic architectures[22].

During our comprehensive immunophenotypic analysis, we noticed two unexpected features. First, we found that immune cells with suppressive functions against anti-tumor cytotoxicity, such as Tregs, MDSCs, and tumor-associated macrophages, are highly enriched in immunologically hot areas. This phenomenon has been reported to occur in many autoimmune diseases and likely reflects the negative feedback mechanism embedded in the systemic nature of immune regulation[40]. Nevertheless, our study suggests that the abundance and composition of immune suppressor cells do not predict the overall suitability of an individual tumor for immunotherapy. Second, we were surprised to observe that, while the TMB and neoantigen loads are moderately associated with local T-cell expansion, they do not associate with local anti-tumor cytotoxicity. These findings are consistent with a recent report that there is no significant correlation between TMB and cytotoxicity in untreated patients[24]. Thus, our results indicate that proliferation and cytotoxicity are two independent parameters for measuring anti-tumor immune activation. T cells can be recruited and activated by local neoantigens to expand, but there still are local suppressive mechanisms to protect tumor cells from killing such as PD-1/PD-L1 axis. Taken together, the disassociation between TMB and cytotoxicity in untreated cohort does not challenge the proof of evidence that TMB could serve as a biomarker and predict clinic outcome from anti-PD-1/PD-L1 treatment in a number of clinic trials[13,14,41].

Our results further reveal that, as related to the immunoregulatory ramifications of somatic mutation in NSCLC, the nature of mutations could play a much more significant role than the sheer number of mutations. In two recent studies, "antigen fitness" was introduced to describe the similarity of neoantigens to other known microbial antigens and their likelihood to be recognized by TCRs. This new parameter is significantly superior to TMB in predicting not only the prognosis of pancreatic cancer patients undergoing standard treatments[25], but also the sensitivity of melanoma and lung cancer patients to anti-CTLA4[42,43] and anti-PD1[13] checkpoint blockade therapies[44]. These studies suggest that neoantigens are not all equal in eliciting TCR recognition. This concept provides a plausible mechanism underlying the limited predictive value of TMB: a large pool of neoantigens could all be weak; conversely, strong antigenicity might be generated by a small pool of high-quality neoantigens.

In addition, the uncoupling of TMB and local cytotoxicity may directly reflect the complex co-selection process between tumor and immune cells. During the prolonged interplay between cancer and the immune system, while mutations elicit adaptive immunity against a tumor, some specific mutations may be critical to reprogram the TME and facilitate their escape from immunosurveillance. This mechanism of immune editing was

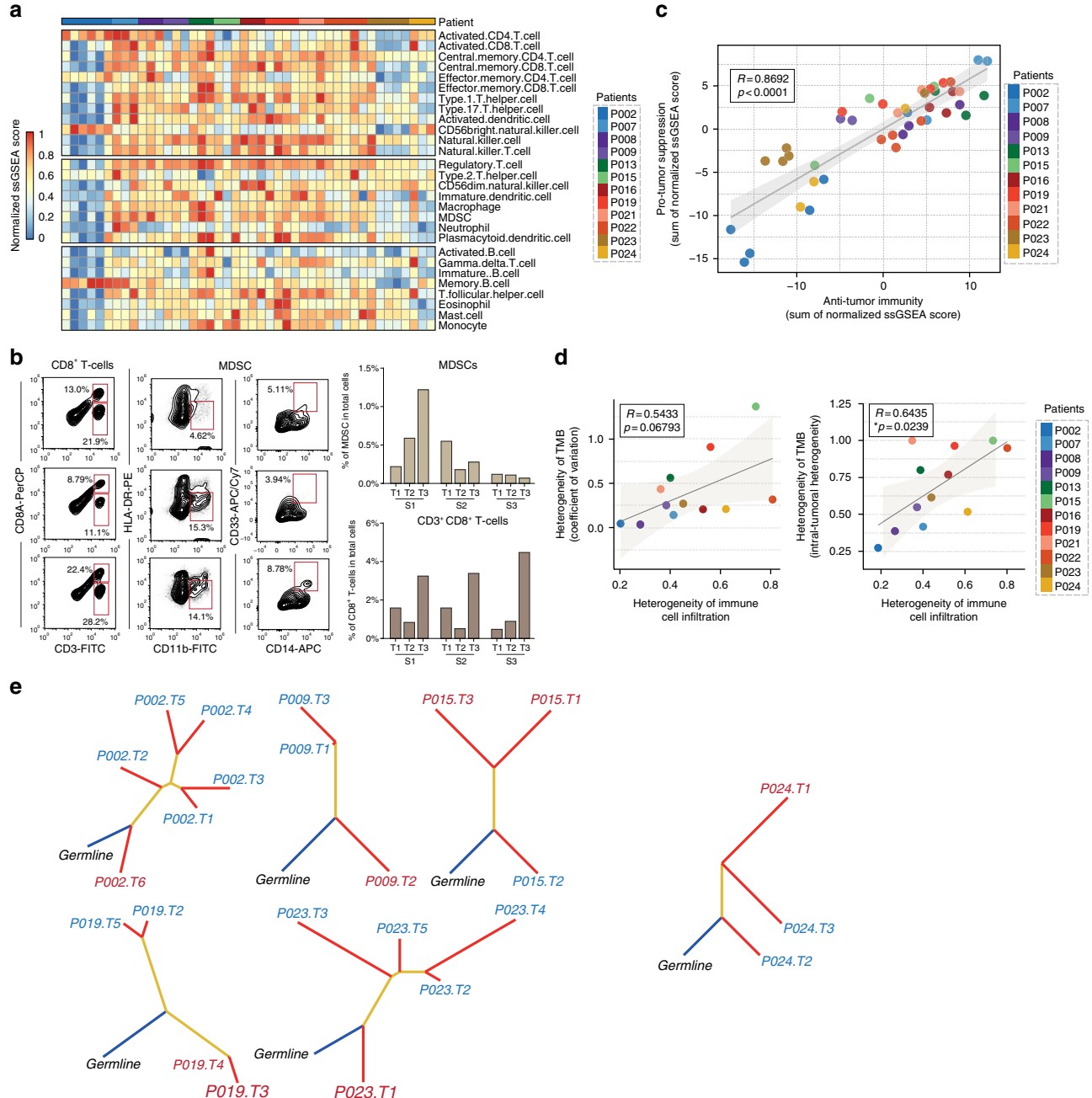

**Fig. 3** Correlation of TMB and immune cell infiltration heterogeneity. **a** Single-sample gene set enrichment analysis identifying the relative infiltration of immune cell populations for 44 NSCLC tumor samples with available RNA-sequencing data. The relative infiltration of each cell type is normalized into a z-score. **b** Validation by flow cytometry. Samples were collected separately from the sequenced batch. **c** Correlation between infiltration of cell types executing anti-tumor immunity (ActCD4, ActCD8, TcmCD4, TcmCD8, TemCD4, TemCD8, Th1, Th17, ActDC, CD56briNK, NK, NKT) and cell types executing pro-tumor, immune suppressive functions (Treg, Th2, CD56dimNK, imDC, TAM, MDSC, Neutrophil, and pDC). R coefficient of Pearson's correlation. The shaded area represents 95% confident interval. **d** Scatterplot showing correlation between heterogeneities of total mutation burden and immune cell infiltration. An average pairwise correlation coefficient was calculated to quantify the divergence of immune cell infiltration, and presented as 1-value for clear visualization. Higher values, heterogeneous immune cell infiltration; lower values, homogeneous immune cell infiltration. The variation in genomic mutations was determined by either intratumoral heterogeneity (ITH) or coefficient of variance (CV). Higher values, heterogeneous; lower values, homogeneous mutation pattern. R coefficient of Pearson's correlation. **e** Phylogenetic trees generated by a parsimony ratchet approach based on the distribution of all detected mutations are shown for patients with heterogeneous immunological status; trunk and branch lengths are proportional to the number of non-synonymous mutations acquired. Red label, samples in hot area; blue label, samples in cold area

demonstrated for somatic mutations associated with antigen presentation and tumor-intrinsic resistance to apoptosis[26]. Therefore, it is conceivable that many mutations may have a direct impact on the TME and thereby regulate the anti-tumor response. Tumor cells with mutations favoring anti-tumor

cytotoxicity constitute an immunologically hot area. However, these tumor cells will be under strong selective pressure, leading to the accumulation of cells with mutations that may be able to switch the TME to immunological coldness. Therefore, intratumoral genetic heterogeneity not only results in phenotypic

diversity and a divergent response to tumor-targeting drugs[45–47], but also the spatial heterogeneity of immune responses.

In summary, this study cautions against the use of overly simplified immune-monitoring strategies to guide immune therapies. It suggests that multi-parameter and multi-locus analysis will aid efforts to assess the territorial and heterogeneous features of immune niches within tumors, resulting in the development of more precise and personalized immune therapies to treat cancers.

## Methods

**Sample collection**. Multi-locus tumor biopsies and clinical data were obtained for 15 patients with NSCLC (4 with squamous cell carcinoma, 10 with non-squamous cell carcinoma, and 1 with not otherwise specified (NOS)); peripheral blood samples were included as germline controls for all cases with whole-exome sequencing. All multi-locus biopsies were obtained as far as possible within the tumor mass during surgery[48]. Resected tumor masses were reviewed by two independent pathologists in Southwest hospital. Detailed clinical characteristics of patients are provided in Supplementary Data. All patient material and clinical information was obtained after informed consent had been received and was approved by the institutional review board of the Southwest Hospital of Third Military Medicine University. DNA and RNA extractions were performed[49]. RNA quality was assessed on a 2100 Bio Analyzer (Agilent Technologies, Santa Clara, CA). Only high-quality DNA and RNA samples were utilized for further sequencing.

**Gene expression profiling**. The mRNA libraries were prepared using the NEB-Next® Ultra™ RNA Library Prep Kit for Illumina® according to the manufacturer's protocol, RNA-seq libraries were paired-end sequenced on an Illumina HiSeq XTEN sequencer (Illumina, San Diego, CA, USA). After removing sequencing reads containing adaptor sequences and low-quality reads, which have too many Ns (>10%) and low-quality bases (>50% bases with quality <5), high-quality paired-end reads were aligned to the human genome (hg19) using HISAT2[50] (v2.0.4). Transcript assembly was performed using StringTie[51] (v1.2.3).

**Whole-exome sequencing**. For each tumor locus and matched germline sample, exome capture was performed using 1–2 μg DNA with the Nimblegen Human All Exome V3 kit, according to the manufacturer's protocol (Nimblegen[52]). Paired-end sequencing was done with an Illumina High-output flow cell kit (300 cycles) on the Illumina HiSeq XTEN platform to a goal of 115-X mean target coverage (detailed coverage information is provided in Supplementary Data). After removing sequencing reads containing adaptor sequences and low-quality reads, which have too many Ns (>10%) and low-quality bases (>50% bases with quality <5), high-quality paired-end reads were aligned to the human reference genome (build hg19, including unknown contigs) using the BWA-MEM (bwa-0.7.15) aligner in default mode. Alignments were then sorted and duplications marked using Samtools (v1.3.1) (http://samtools.sourceforge.net) and Picard (v2.6) (https://broadinstitute.github.io/picard/), respectively. BAM files were indel-realigned and base quality scores were recalibrated according to GATK best practices (https://software.broadinstitute.org/gatk/). Using WES data, tumor cell purities and ploidies were calculated based on calls made by the Sequenza[53] R package. A default parameter (~w 50) was used to run the analysis. Six samples from three patients were re-sequenced to assess WES reproducibility.

**Identification of somatic mutations**. MuTect2 (http://www.broadinstitute.org/cancer/cga/mutect) was used to identify somatic single-nucleotide variants and small insertions or deletions with default parameters based on paired alignment files (tumor and matched germline). Mutations were excluded step by step to select candidates of potential interest. First, non-silent variants, including missense, nonsense, frameshift, and splice-site variants, were selected. Second, high-confidence variants whose variant allele fraction was greater than 0.05, or coverage at least 5×, were selected. Third, rare variants with frequencies less than 0.005 in all databases (ExAC, ESP6500, dbSNP, 1000G) were selected.

**High-throughput sequencing of TCR β-chain**. To generate the template library for the T-cell repertoire, a multiplex PCR system was designed to amplify the third complementarity-determining region (CDR3) of TCRB[54]. Briefly, 600 ng gDNA for each sample was amplified using two rounds of PCR. During the first round, 10 cycles were used to amplify the CDR3 fragments using 32 forward primers of V genes, and 13 reverse primers of J genes with a Multiplex PCR Kit (QIAGEN, Germany). Primers were designed to acquire maximum coverage of a heterogeneous set of target sequences of V and J families with a minimal PCR bias. Primer sequences were filed as part of a Chinese patent (CN105087789A). In the second round, PCR was performed using Illumina universal primers with a Phusion® High-Fidelity PCR Kit (New England Biolabs, USA). Paired-end sequencing of samples was carried out with a read length of 151 bp using the Illumina HiSeq

XTEN platform. Raw reads were processed and analyzed using the following procedure: (1) removing sequencing reads which do not contain the primers for multi-PCR using Cutadapt (https://github.com/marcelm/cutadapt); (2) merging the remaining high-quality pair-end reads to obtain contigs by Pear;[54] and (3) spotting of the CDR3 region using MiXCR[55] (https://github.com/milaboratory/mixcr) with default parameters.

Diversity and clonality were used to characterize features of the immune repertoire. Diversity of the TCR repertoire was calculated based on the Shannon–Wiener index (Shannon entropy), which is a function of both the relative number of clonotypes present and the relative abundance or distribution of each clonotype[19]. Clonality was defined as 1–(Shannon entropy index)/ln(number of productive unique sequences)[8].

**Characterization of HLA molecules and neoantigens**. For each subject, the 4-digit HLA type was inferred using Optitype[56], which uses a normal germline bam file as input. HLA typing was further confirmed using the HLA-VBseq[57] tool. Chromosome 6 was extracted from the WES data and aligned to the HLA genome reference from the IMGT/HLA database. Identified non-silent mutations from WES were used to generate a comprehensive list of peptides 8–11 amino acids in length with the mutated amino acid represented in each possible position. The binding affinity of every mutant peptide and its corresponding wild-type peptide to the patient's germline HLA alleles were predicted using netMHCpan-3.0[58]. Predicted neoantigens in correlation analysis were identified as those with a predicted binding strength of <50 nM and mutant peptide binding affinity less than 70% of wild-type binding affinity.

**Identification of clonal and subclonal somatic mutation**. Two methods were used to identify clonal and subclonal somatic mutations[22,27,59]. One, developed by Blakely et al.[27], defines the frequency of each somatic mutational allele (minor allele frequency (MAF)) after normalization by ploidy (extracted from Sequenza analysis). The value of each MAF is then divided by the maximal value of all MAFs to reach the normalized MAF (nMAF). If nMAF ≥ 0.2, this mutation is defined as a clonal somatic mutation, whereas the nMAF value for subclonal somatic mutations is <0.2. Alternatively, McGranahan et al.[22] define clonal mutations as mutations that present in all collected loci after multi-sampling a patient; conversely, any mutations that only present in a subset of loci are defined as subclonal mutations.

**Phylogenetic tree construction**. All non-synonymous somatic mutations were utilized for phylogenetic tree construction. Germline mutations were obtained from paired peripheral blood samples. Trees were constructed using binary presence/absence matrices assembled from the locus distribution of variants within the tumor. The R Bioconductor package was used to generate unrooted trees by the parsimony ratchet method[54]. Branch lengths were determined using the acctran function.

**Immune cell infiltration**. The ssGSEA[29] was introduced to quantify the relative infiltration of 28 immune cell types in the tumor microenvironment. Feature gene panels for each immune cell type were obtained from a recent publication[17]. The relative abundance of each immune cell type was represented by an enrichment score in ssGSEA analysis. The ssGSEA score was normalized to unity distribution, for which zero is the minimal and one is the maximal score for each immune cell type. The bio-similarity of the immune cell filtration was estimated by multi-dimensional scaling (MDS) and a Gaussian fitting model.

**Machine learning and visualization of immune map**. "Immune maps" were generated from a random forest machine learning algorithm using 278 input variables (Supplementary Data). Briefly, a random process was performed with 100 iterations. For each round, two-thirds of samples were randomly selected as the discovery set, and one-third of samples were assigned as the validation set. Randomness was determined using a standard random number generator program in the R package. A prediction model (decision tree) was calculated based on each discovery/validation set. After 100 iterations, the resulting 100 decision trees were incorporated into one proximity matrix to minimize any artificial or random bias. RandomForest R package was used to run this analysis. MDS was used to assign a spatial location to each sample according to the proximity matrix from machine learning and 278 dimensions of defined immune-related features.

**Immunohistochemical staining of PD-L1**. PD-L1 expression was analyzed using formalin-fixed, paraffin-embedded biopsy samples by the ShuWenGuanZhi Company (Huzhou, Zhejiang, China) in accordance with all American Society of Pathology CAP and ISO15189 quality management standards.

**Tissue dissection and flow cytometry**. Tumor tissues were processed into a single-cell suspension. All antibodies were purchased from BioLegend. The single-cell suspension was stained with anti-human CD45-PacificBlue (clone HI30), anti-human CD33-APC/Cy7 (clone P67.6), anti-human CD14-APC (clone 63D3), anti-human CD3-FITC (clone HIT3a), anti-human CD8-PerCP (clone HIT8a), and

anti-human HLA-DR-PE (clone L243) antibodies, and analyzed using a fluorescence-activated cell sorter (FACS) (FACSAria II; BD Biosciences).

## Data availability

The gene expression profiling data supporting the conclusions of this project have been deposited publicly to the Genome Expression Omnibus (GEO) under accession code GSE112996. Whole-exome sequencing and T-cell repertoire sequencing data have been deposited in Sequence Read Archive (SRA) under accession code PRJNA493023 and PRJNA506151. All other data are available within the Article and its Supplementary Information or from the corresponding authors upon reasonable request.

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

## Acknowledgements

We acknowledge support by the National Nature Science Foundation of China (81222031) to B.Z., and Beau Biden Cancer Moonshot Initiative from the NIH (R33CA225328) to Q.-J.L.

## Author contributions

B.Z. and Q.-J.L. designed the experiments. W.W. and Q.J. prepared the samples. Y.W., H. S., Y.G., X.X., L.Y., J.H., and X.Y. acquired the data. Q.J., C.S., Z.G., J.C., J.H., and H.S. contributed to the data analysis. Q.J., P.B.A., Q.-J.L and B.Z. drafted the manuscript. All authors critically reviewed the paper.

## Additional information

**Competing interests:** The authors declare no competing interests.

