## [Peer Review File · Nature Communications]

Reviewers' Comments:

Reviewer #1:

Remarks to the Author:

Review of Jia et al.

Jia et al. examine intratumoral heterogeneity by performing WES, transcriptome profiling, and TCR profiling in different parts of tumors in NSCLC patients. They find that the immune microenvironment is compartmentalized. They claim that while local TMB is associated with local T cell clonal expansion, it did not predict local anti-tumor cytotoxicity. There are major issues with this paper.

Comments:

1. There are serious logical problems with the interpretation of that data. The authors state many times that there is evidence that challenge the association between TMB and anti-tumor response. In fact, it is well known that tmb is a predictive marker of immunotherapy response, although this is a general association, not an absolute rule. TMB is not necessarily a marker of anti-tumor response or cytotoxicity in the absence of immunotherapy treatment. This is exactly because immune checkpoints are operative. These concepts are blurred throughout the paper and make the logic and premise of the entire manuscript faulty. The authors should strive to be clear on what they mean and refine their interpretations.

2. "It also calls into question whether total mutation burden or neoantigen load is decisive for predicting immune "readiness" for immunotherapy."

It is unclear what the authors mean here. What is immune "readiness"? this statement is quite misleading. The studies on TMB and MSI clearly show that TMB predicts with IO agent response. The authors should temper their statements as its unlikely their small 15 patient study would have much power to call into question phase III studies and the cumulative data from thousands of patients. Language is important here and the authors should be more mindful.

3. Did the authors perform orthogonal validation sequencing? If so, this needs to be presented.

4. It is not clear how the authors selected the regions to be sampled? Were the sites examined previously by a pathologist to ensure that necrosis and fibrosis was accounted for?

If so, the authors need to show this data for transparency. Otherwise, it is unclear what they are sequencing.

5. Fig 1B-D. The x axis is unreadable. Authors need to show this data somewhere so its directly readable. The authors need to make the spatial relationships between these data clear in terms of tumor location.

6. Inflamed signatures and the lack of correlation with TMB in the untreated setting has already been explored by others, such as Spranger et al. PNAS (PMID: 27837020). The finding that spontaneous immune activity does not correlate with TMB is thus not novel. Furthermore, it is already known that immune enriched microenvironments only loosely correlate with response to immune checkpoint inhibitors. This paper adds nothing novel on this front.

7. There are issues with the machine learning work. It seems that the authors have selected specific genes to include. How did the authors control for bias here for input variable selection?

8. The authors have not controlled for local immune checkpoint expression or immune suppressive immune subsets (ie. MDSC + others) that may prevent neoantigens from being targeted. The analysis of association they performed did not account for confounders.

9. The authors did not validate their immune map with pathologic assessment but relied on only gene expression level data. Furthermore, they have no idea which cellular compartment these genes come from. The immune map analysis is at best preliminary. Flow destroys the spatial information that could be gained.

10. It is well known that immune recognition may be influenced by mutational clonality. When clonality is taken into consideration (ie. clonal vs subclonal mutation burden), does immune activity correlate with higher burden? Fig 3 begins to address this but does not parse out this important analysis.

Reviewer #2:

Remarks to the Author:

General comments

The manuscript by Jia et al. describes a study on the spatial heterogeneity of the genetic and immune profiles in NSCLC. The authors analyzed multiple biopsy samples from 15 patients and identified immune heterogeneity within patients. Furthermore, the results indicate that the mutational burden was not associated with immune cytotoxicity.

While intratumoral genetic heterogeneity has been well studied, the heterogeneity of the tumor immunity has not been comprehensively characterized. Thus, the findings of the study are of potential interest to a wider audience. However, there are several issues that need to be addressed. First, the figures and the figure labels are very small and extremely hard to read. One has to zoom in 400% in order to read the labels. All figures should be redone and enlarged. Second, the method section is too brief and buried in the supplementary material. It is difficult to evaluate the study given insufficient information (see also specific comments). Third, the raw data was not deposited in an appropriate database and hence, the results of the analyses cannot be reproduced. It is an imperative to deposit data in a public repository. Notably, there are publicly available repositories that can be used to deposit data and access the data –also with restrictions due to privacy limitations. For example, data deposited in GDC/dbGAP have a controlled access and clearly specified procedures to access the data. And fourth, many of the statements are either not correct or even speculative and the authors should rewrite the manuscript and carefully describe the findings.

Specific comments

The use of the word “compartmentalized” in the title and throughout the manuscript is not optimal, as it would refer at clearly separate/identifiable areas or structures of the tumors. “Spatial heterogeneity” would be more appropriate to describe the molecular and immunological diversity.

Different ecological measures of diversity, including the Shannon’s entropy, inverse Simpson and complementary Simpson indices, can be applied to describe T cell repertoires. However, there is no “Shannon clonality index” as reported in the manuscript. Shannon entropy-based clonality index is more appropriate. Both indices should be described in the methods section.

L. 157-164: The lack of thorough explanation in the Methods section hampers the evaluation of some of the analyses performed. For instance, it is not clear how the “immune map” was built. From caption Figure 2 caption it seems that MDS was used. If so, the immune map would be far from being robust (as claimed at L. 178), as the location of each sample in the immune map would depend on the set of samples considered in the MDS analysis.

L. 208-209: Please clarify how the “distinct phenotypic nature of different mutations may directly modify the immune response”.

L. 216-218: The authors refer to immunologically hot tumors, but it is not clear from the figure how were these tumors defined. Furthermore, the data is not sufficient to state that cells are executing anti-tumor immunity or pro-tumor suppression. It would be better to name the cell types.

L. 218-222: Can the correlation of the immune-cell NES be biased due to tumor purity?

Figures

Figure 1:

What are the shaded area and the line in Figures 1E-F?

The "Machine Learning" panel of Figure 1A seems to have been copied and pasted from Hackl et al. (Nature Reviews Genetics, 2016), with obvious copyright issues.

Replace in Figure 1A "foci" with "loci".

Figure 1C: predicted neoantigens should be filtered for expressed ones (using RNA-seq data). A figure with the neoantigen burden (as 1E and 1F) would be helpful.

Figure 2:

Instead of dichotomizing the cytolytic activity into two classes, it would be better to plot it as continuous variable, like for gene expression levels.

Figure 3:

Figure 3A which variable was plotted (NES?).. The legend should also explain what are the shaded area and the line of Figure 3C.

Reviewer #3:

Remarks to the Author:

NCOMMS-17-33520

The authors analyzed different regions of lung cancer for somatic mutations, TCR-beta repertoire and immune signature by DNA-seq and RNA-seq. By standard techniques (NetMHC), they predicted neoepitopes and tried to correlate neoepitopes with T cell clonal expansion and a signature of immune cytotoxicity. They found significant heterogeneity in different tumor regions, T cell expansion was associated with mutation burden but not with an immune cytotoxicity signature. These data contrast some prominently published papers, in which the immune cytotoxicity signature was correlated with decreased predicted neoepitopes. These earlier studies were performed with whole tumor tissue and, therefore, the current manuscript is important, since it demonstrates that tumor heterogeneity, at least of tumors with high mutational burden, introduces a complexity, which when ignored can lead to substantial false-positive results.

A general critique on the current and all related studies, of which almost weekly appear new ones, is neoepitope prediction. None of these studies in human cancer demonstrated that a predicted neoepitope is indeed a neoepitope. Current algorithms predict the peptide-MHC binding affinity (but unlikely for all MHC I alleles) but cannot accurately predict proteasomal processing and not at all post-proteasomal trimming and whether sufficient peptide-MHC complexes reach the cell surface. With vaccinia virus as model, it was shown that only a minor fraction of HLA-A2 predicted epitopes were actually immunogenic in HLA-A2 transgenic mice (Assarsson et al. J Immunol 2007, 178: 7890–7901). No reason to assume that it is different for cancer neoantigens. Thus, it is unclear how large the number of wrongly predicted neoepitopes is. A cut-off of IC50 of 500 nM is quite low, unclear whether such epitopes can elicit a T cell response in the cancer context. How do the results look like, if cut-off is 50 or 5 nM?

That the T cells in the lung tumors are neoepitope-specific is possible but speculative and relies on

data from few melanoma patients treated successfully with TILs.

The terms tumor immunity, immunogenic or antigenic are poorly speculative. Immunity means "protection" but the T cells do not prevent tumor progression. "Reactivity" seems a more appropriate term (Nat Rev Cancer 12: 307-313, 2012). The authors have no indication that the expanded T cells ever were functional.

It is not possible for this reviewer to go in detail through this vast amount of omics data and judge the bioinformatics. However, all conclusions of the current manuscript depend not only on accurate prediction of neoepitopes (see above) but also on accurate determination of the HLA class I haplotype. It seems that Optitype is not 100% accurate (Journal of Human Genetics (2017) 62, 397–405).

For better judging the data, a Table with all mutations, predicted IC50 and RNAseq reads would be helpful.

Fig 1B-D are not understandable.

Figure labels are too small.

Lines 123-138: Lavin et al., 2017, Cell 169, 750–765 might be relevant to be cited.

Suppl Fig S3 should be labeled.

Reading the manuscript was difficult, since some Suppl Tables were wrongly labeled, e.g. Suppl Table 5 was missing.

In conclusion, bearing the above critiques in mind, the manuscript is interesting, because it puts into question current popular views and illustrates the problem with this type of analysis.

Response to Reviewers' comments:

We sincerely thank all three reviewers for their insightful and constructive suggestions. To address reviewer concerns and strengthen our study, we have now performed additional analyses and made the following modifications: 1) we assessed the clonality of somatic mutations, clonal versus subclonal, and found that clonal somatic mutations are also loosely associated with an inflamed immune signature. This is similar to what we observed for the total somatic mutational burden; 2) we calculated the purity of tumor cells in our tumor samples and found that ssGSEA-estimated immune cell infiltration is independent of tumor cell purity, suggesting the ssGSEA score is not significantly biased in the sampling; 3) we re-analyzed HLA typing using the HLA-VBSeq tool, which produced identical results as our previous analysis using the Optitype algorithm; 4) we elaborated our method section, including clarifications and detailed descriptions for our key bioinformatics analyses, to enhance our data clarity. Based on these new results and general comments related to data interpretation from the reviewers, we have thoroughly revised the manuscript.

In addition, we would like to clarify two of the major conclusions of our manuscript. First, our analysis indeed challenges the idea that the tumor mutational burden (TMB) or its associated neoantigen load is the best biomarker to predict local anti-tumor immune responses. Instead, we believe that the nature, or immunological “quality”, of mutations is as or more important than sheer quantity of mutations. This idea is supported by a recently published “antigen fitness” model^{1,2} from other groups that had previously advocated for the importance of TMB. These studies utilize new patient cohorts or revisit previously attained clinical data to conclude that the antigenicity of mutations has superior predictive value than mutational abundance. We do not have evidence, and indeed have not claimed, that TMB is a failed biomarker for checkpoint blockade prognosis. However, it must be acknowledged that new results from our lab and others indicates that tumor antigenicity has better association with anti-PD1 and anti-CTLA4 efficacy compared to TMB.

The second critical issue we would like to address here is that our study heavily emphasizes immune microenvironmental heterogeneity within individual tumors. Our discovery that immune reactions are spatially heterogeneous within the TME does challenge the value of biomarkers obtained from single locus biopsy. Indeed, the unexpectedly high immune heterogeneity we demonstrate here may directly reflect the limitations of biomarkers currently used for immunotherapy prognostic prediction.

To reflect these changes and to clear our message, we changed the title of this manuscript to “Local Mutational Diversity Drives Intratumoral Immune

Heterogeneity in Non-Small Cell Lung Cancer”. Please find enclosed a point-by-point response to all issues raised by the reviewers. Any changes to the revised manuscript including those specifically addressing reviewer concerns have been highlighted in the manuscript file.

RESPONSES TO REVIEWERS

We would like to express our sincere thanks to all three reviewers for their critical and constructive comments. To address reviewer concerns we have performed substantial additional analyses, which we feel have helped us clarify important issues and significantly improve the manuscript. Below we respond point-by-point to each of the reviewer comments.

Responses to Reviewer #1

1. There are serious logical problems with the interpretation of that data. The authors state many times that there is evidence that challenge the association between TMB and anti-tumor response. In fact, it is well known that TMB is a predictive marker of immunotherapy response, although this is a general association, not an absolute rule. TMB is not necessarily a marker of anti-tumor response or cytotoxicity in the absence of immunotherapy treatment. This is exactly because immune checkpoints are operative. These concepts are blurred throughout the paper and make the logic and premise of the entire manuscript faulty. The authors should strive to be clear on what they mean and refine their interpretations.

We thank the reviewer for pointing out that our manuscript's message needs clarification. We are indeed challenging the idea that TMB directly associates with intrinsic anti-tumor responses. We do not have evidence that contradicts the established clinical association between TMB and the efficacy of checkpoint blockade immunotherapy. However, the predictive value of TMB has been challenged by two recent reports^{1,2}, including one from previous advocates of the importance of TMB³. In these studies of melanoma⁴ and pancreatic cancer¹ patient cohorts not undergoing immunotherapy, TMB failed to predict immune responses or prognosis. Moreover, when three clinical trials (two with anti-CTLA4 against melanoma^{4,5} and one with anti-PD1 against NSCLC³) were revisited, the general consensus was that neoantigen quality, which represents the antigenicity of mutations, has superior predictive value compared to neoantigen quantity (TMB)². Based on these recent studies, it is reasonable to conclude that, although TMB is currently the best predictive marker of checkpoint blockade responses, its prognostic value is not optimal. Our conclusions agree with those of others^{1,2,4-7} to emphasize that : a) the nature or quality of mutations is a critical determinant of immunogenicity; b) an ideal biomarker to describe intratumoral immune responses depends on more sophisticated analyses, which include TMB as one parameter.

To clarify this message, we have revised our manuscript thoroughly such that it now specifically focuses on our finding that the TMB is not directly related to the local anti-tumor response in the absence of immunotherapy. We also wrote an additional paragraph to discuss the recent “antigen fitness” model, which provides further contextual support for our conclusions.

2. “It also calls into question whether total mutation burden or neoantigen load is decisive for predicting immune “readiness” for immunotherapy.” It is unclear what the authors mean here. What is immune “readiness”? this statement is quite misleading. The studies on TMB and MSI clearly show that TMB predicts with IO agent response. The authors should temper their statements as its unlikely their small 15 patient study would have much power to call into question phase III studies and the cumulative data from thousands of patients. Language is important here and the authors should be more mindful.

We changed “readiness” to “suitability” for immunotherapy. Moreover, we have elaborated on our views, in the context of recent progress made in this field, about the predictive value of TMB for local immune reactivity and checkpoint blockade efficacy as described above. We agree that currently TMB is still the best biomarker for IO reagent response. However, it should be noted that only ~50% of NCCLC patients with high TMB receive benefit from such immunotherapy⁸. Therefore, it remains an urgent need to improve upon this biomarker and the first step to do so is to understand the limitations of TMB and single locus biopsy; we feel that this insight is one of the main values of our manuscript. Although it is unaffordable to perform comprehensive immunogenomics analysis on thousands of patient samples, we agree that our cohort size is small and we need to use more precise language to present our data and interpretations; we have attempted to improve in this regard throughout the revised manuscript.

3. Did the authors perform orthogonal validation sequencing? If so, this needs to be presented.

We have not performed orthogonal validation sequencing for this project. Our sequencing data were generated according to the standard operating procedures of the clinical diagnosis lab Geneplus Corp, which also provides sequencing services in genetic screening and molecular diagnosis for patients. The sequencing accuracy and reproducibility of GenePlus have been approved by China’s Food and Drug Administration and inspected and certified by the National Health Commission of

China annually. The service certifications for Geneplus in cancer-related diagnosis can be found at <http://www.geneplus.org.cn/trans/toAboutUs>.

4. It is not clear how the authors selected the regions to be sampled? Were the sites examined previously by a pathologist to ensure that necrosis and fibrosis was accounted for? If so, the authors need to show this data for transparency. Otherwise, it is unclear what they are sequencing.

In this study, all sampled biopsies were obtained *in situ* during tumor resection. Biopsies were taken from regions as far apart as possible by the surgeon inside the invasive marginal tissue. All tissues were reviewed by two independent pathologists in the Southwest Hospital to make sure the majority of sample was tumor tissue. We now include a description of this procedure in the revised Materials and Methods section.

In addition, per Reviewer #2's suggestion, the purity of tumor cells in the analyzed samples were assessed using the Sequenza software package⁹. This analysis revealed a mean tumor cell purity above 40% (ranging from 21% to 82%), which is comparable to other similar and recent publications for melanoma (54% on average, ranging from 14% to 95%)¹⁰, prostate cancer (34% on average, ranging from 23% to 79%)¹¹, and neuroblastoma (78% on average, ranging from 22% to 98%)¹². This QC data set is now presented as **Supplementary Table 2**. In addition, we plotted the tumor cell purity against the normalized enrichment score (NES) of each infiltrating immune cell population (**Response Figure 2**). We found that tumor loci with lower purity are in general those with higher NES for neutrophils, macrophages and activated/differentiated T cells. This suggests that immune cell infiltration may contribute significantly to lower tumor tissue purity scores.

5. Fig 1B-D. The x axis is unreadable. Authors need to show this data somewhere so it is directly readable. The authors need to make the spatial relationships between these data clear in terms of tumor location.

We have removed the x-axis label in the original Figure 1B-D and presented high-resolution images as **Supplementary Figure 2**. To present the spatial relationships, all samples from the same patient are now annotated in coded colors.

6. Inflamed signatures and the lack of correlation with TMB in the untreated setting has already been explored by others, such as Spranger et al. PNAS (PMID: 27837020). The finding that spontaneous immune activity does not correlate with TMB is thus not novel. Furthermore, it is already known that immune enriched microenvironments only loosely correlate with response to immune checkpoint inhibitors. This paper adds nothing novel on this front.

The main novel findings of our manuscript are as follows: 1) we provide direct evidence that intratumoral immune cytotoxicity is highly heterogeneous in different regions of a single tumor; 2) we show that the TMB is moderately associated with local T cell clonal expansion, but cannot predict whether that locus is inflamed or associated with PD1 ligand expression. Furthermore, our data also suggest that the nature of mutations, rather than the number of mutations, may be more critical for reprogramming the immune microenvironment; 3) we show that immune heterogeneity is associated with the complexity of immune cell infiltration.

7. There are issues with the machine learning work. It seems that the authors have selected specific genes to include. How did the authors control for bias here for input variable selection?

The essential hypothesis for our immunogenomics analysis is that a broader knowledge base can enhance the accuracy of prediction. Armed with this broad base, we let machine learning help us to reduce complexity and make simple categorization. It is true that our input variables were selective but they are selected based on previous findings and reasonable assumptions. To our knowledge, in comparison to other recently published algorithms⁷, our immune map is built on the highest number of variables. We included the expression of 31 immunoregulatory genes (e.g., ICOS, IDO1, PDCD1, TIGIT, LAG3, etc.) and the abundance of 28 subpopulations of infiltrating immune cells (e.g., activated CD8⁺ T-cells, MDSC, Tregs, etc.), which are variables commonly selected by other analytic programs^{7,13}. In addition, we took neoantigen loads and T cell repertoire clonality into consideration. Furthermore, since signal transduction provides the fundamental mechanism underlying tumor cell growth and metabolism (e.g., JAK, Wnt), immune regulation (e.g., STATs, NFκB), and microenvironment formation (e.g., proteins involved in angiogenesis and hypoxia), we calculated 217 enrichment scores representing 217 signal transduction pathways from BIOCARTA database¹⁴. Taken together, this shaped our selection strategy for variable inputs.

8. The authors have not controlled for local immune checkpoint expression or immune suppressive immune subsets (ie. MDSC + others) that may prevent neoantigens from being targeted. The analysis of association they performed did not account for confounders.

As presented above, the relevant abundance of suppressive immune populations, such as Tregs, Th2 cells, CD56^{dim} NK cells, immature DCs, plasmacytoid DCs, tumor associated macrophages, neutrophils, and MDSCs, have been taken into account in our algorithm. Also, our machine learning program included well-established immune checkpoint molecules, such as ICOS, IDO1, PDCD1, CTLA4, TIGIT, LAG3, HAVCR2, CD274, and HHLA2. These parameters are summarized in **Supplementary Table S9**.

9. The authors did not validate their immune map with pathologic assessment but relied on only gene expression level data. Furthermore, they have no idea which cellular compartment these genes come from. The immune map analysis is at best preliminary. Flow destroys the spatial information that could be gained.

In recent years, new algorithms deconvoluting gene expression data represent one of most exciting bioinformatics achievements. Dissecting out immune cell subset frequencies from tissue expression profiles is an efficient way to assess tissue microenvironment¹⁵. The prototype algorithm, which uses gene set enrichment analysis (GSEA) to dissect cell populations, was first established by Barbie et al., in 2009¹⁶. In 2010, the first signature gene set was developed to dissect immune cell population¹⁷. Following these successes, numerous subset-specific signatures have been designed and validation experiments have been performed to improve the accuracy of enumeration¹⁸⁻²¹. As of today, this is a mature tool and frequently used in tumor microenvironmental analysis^{7,22-24}. The subset-specific gene signature adopted in our assays has been validated in previous publications^{7,22,23}. We agree with Reviewer #1 that the limitation of this analysis is the loss of spatial information. **Figure 3B** represents an effort to validate the computation-aided prediction.

10. It is well known that immune recognition may be influenced by mutational clonality. When clonality is taken into consideration (ie. clonal vs subclonal mutation burden), does immune activity correlate with higher burden? Fig 3 begins to address this but does not parse out this important analysis.

We thank Reviewer #1 for this insightful suggestion. Indeed, higher clonal mutational burden and low subclonal mutational heterogeneity is associated with superior prognosis in immunotherapy-naïve NSCLC cohort and clinically benefited group in checkpoint blockade therapies²⁵. To exclude subclonal mutations as a confounding factor, we dissected clonal mutational burdens using two independent methods^{25,26}. One is developed by Blakely et al²⁶, which defines frequency of each somatic mutational allele (MAF) after normalization by the ploidy (extracted from Sequenza analysis). The value of each MAF for each allele is then divided by the maximal value of all MAFs to reach the normalized MAF (nMAF). If nMAF ≥ 0.2 , this mutation is defined as the clonal somatic mutation; and, the nMAF value for subclonal somatic mutation is < 0.2 . Alternatively, McGranahan et al.²⁵ defines clonal mutation as a mutation that presents in all collected loci after multi-sampling a patient; and, any mutations that only presents in the subset of loci is defined as subclonal mutation. As shown in **Response Figure 1**, with both methods, subtraction of subclonal mutations fails to improve the correlation between TMB and local immune cytotoxicity. We now present this data as **Supplementary Figure 4**.

Response Figure 1 (also new Supplementary Figure 4). Correlation between clonal mutational loads and local immune cytotoxicity. Clonal mutations are defined as “ubiquitous and truncal” mutations as previously described by Blakely et al.,²⁶ (**A, B**) or McGranahan et al.,²⁵ (**C, D**) in contrast with subclonal mutations, which are only identified in a fraction of tumor cells. After subtraction of subclonal mutations, the clonal mutational burden was plotted against INF- γ expression or cytolytic activity, a parameter comprised of both PRF1 and GZMA expression. R, coefficient of Pearson correlation. Shaded areas represent the 95% confidence interval of fitting.

Response to Reviewer #2

General comments

The manuscript by Jia et al. describes a study on the spatial heterogeneity of the genetic and immune profiles in NSCLC. The authors analyzed multiple biopsy samples from 15 patients and identified immune heterogeneity within patients. Furthermore, the results indicate that the mutational burden was not associated with immune cytotoxicity.

While intratumoral genetic heterogeneity has been well studied, the heterogeneity of the tumor immunity has not been comprehensively characterized. Thus, the findings of the study are of potential interest to a wider audience.

We appreciate Reviewer #2's comments regarding the novelty and significance of our manuscript.

However, there are several issues that need to be addressed. First, the figures and the figure labels are very small and extremely hard to read. One has to zoom in 400% in order to read the labels. All figures should be redone and enlarged.

We apologize for this unsatisfactory presentation. To address this, we have now enlarged all figures such that their labels are legible.

Second, the method section is too brief and buried in the supplementary material. It is difficult to evaluate the study given insufficient information (see also specific comments).

Per Reviewer #2's suggestion, we now provide additional details in the Materials and Methods section.

Third, the raw data was not deposited in an appropriate database and hence, the results of the analyses cannot be reproduced. It is an imperative to deposit data in a public repository. Notably, there are publicly available repositories that can be used to deposit data and access the data – also with restrictions due to privacy limitations. For example, data deposited in GDC/dbGAP have a controlled access and clearly specified procedures to access the data.

We thank Reviewer #2 for raising this important expectation. We have now uploaded all raw data to the Gene Expression Omnibus (GEO) database. Because this is a large amount of data, the verification steps at GEO are slower than we expected. As of today, we have only received an accession number for the RNASeq dataset, which is GSE112996. We anticipate that the accession numbers for the TCR Rep-seq and WES datasets will be available in the coming weeks.

And fourth, many of the statements are either not correct or even speculative and the authors should rewrite the manuscript and carefully describe the findings.

We now realize based on these and similar concerns raised by Reviewer #1 that we did not deliver a clear message in our initial submission. In an effort to clarify our findings and interpretations, we have now substantially revised the manuscript based on the constructive and insightful comments from all three reviewers. Many of these modifications are described in detail in our response to Reviewer #1's specific comments above.

Specific comments

The use of the word “compartmentalized” in the title and throughout the manuscript is not optimal, as it would refer at clearly separate/identifiable areas or structures of the tumors. “Spatial heterogeneity” would be more appropriate to describe the molecular and immunological diversity.

We like this term “spatial heterogeneity” very much and have adopted it to replace “compartmentalized” throughout the manuscript. Following this suggestion, we also

changed the title of this manuscript to “Local Mutational Diversity Drives Intratumoral Immune Heterogeneity in Non-Small Cell Lung Cancer”.

Different ecological measures of diversity, including the Shannon’s entropy, inverse Simpson and complementary Simpson indices, can be applied to describe T cell repertoires. However, there is no “Shannon clonality index” as reported in the manuscript. Shannon entropy-based clonality index is more appropriate. Both indices should be described in the methods section.

Reviewer #2 is correct on this issue and we apologize for our inappropriate use of this phrase. The parameter presented in Figure 1 is in fact the Shannon entropy-based clonality index. This has been corrected both within the figure as well as in the Materials and Methods section describing our computational process.

L. 157-164: The lack of thorough explanation in the Methods section hampers the evaluation of some of the analyses performed. For instance, it is not clear how the “immune map” was built. From caption Figure 2 caption it seems that MDS was used. If so, the immune map would be far from being robust (as claimed at L. 178), as the location of each sample in the immune map would depend on the set of samples considered in the MDS analysis.

Reviewer #2 is correct that we used MDS to construct the immune map. The three most commonly used tools for dimension reduction are PCA (principal component analysis), MDS (multi-dimensional scaling) and *t*-SNE (*t*-distributed stochastic neighbor embedding). Generally, *t*-SNE is the most robust among the three while PCA is the least robust. However, *t*-SNE is only suitable when analyzing a large number of samples; we do not think our sample size is big enough for it. We chose MDS mainly based on sample size.

Using MDS, we assigned a spatial location for each sample according to 278 dimensions of defined immune-related features. We utilized the random forest machine learning strategy and performed 100 iterations. For each round, 2/3 of samples were randomly selected as the discovery set, and 1/3 samples were assigned as validation set. This randomness was determined using a standard random number generator program in the R package. A prediction model (decision tree) was calculated based on each discovery/validation set. After 100 iterations, the resulting 100 decision trees were incorporated into one proximity matrix to minimize the

artificial or random bias. It is true that the resolution of our map is determined by the dynamic range provided by our data set. However, this method indeed provided a relatively more robust separation than other assays tested in our study. Nevertheless, we understand Reviewer #2's concern and no longer describe the immune map as "robust" in the manuscript. We have also revised the Materials and Methods section to include a description of these analytic procedures.

L. 208-209: Please clarify how the "distinct phenotypic nature of different mutations may directly modify the immune response".

We have modified this sentence as "*different mutations may play distinct roles in modifying the immune response*²⁷".

L. 216-218: The authors refer to immunologically hot tumors, but it is not clear from the figure how were these tumors defined. Furthermore, the data is not sufficient to state that cells are executing anti-tumor immunity or pro-tumor suppression. It would be better to name the cell types.

We have now labeled Figure 2A to more clearly indicate the regions of the immune map designated as immunologically "hot" and "cold" microenvironments. We also now provide additional details in the figure legend to facilitate comprehension. In L. 216-218, per reviewer's suggestions, we list the cell types executing anti-tumor immunity or delivering pro-tumor suppression in the revised manuscript. For the immune map, the regional designation of tumor loci for their immunological phenotype is impacted by local immune cell infiltration. However, abundance of infiltrations is only a fraction of factors taking in consideration to generate immune map. Therefore, we cannot name specific cell types on the immune map.

L. 218-222: Can the correlation of the immune-cell NES be biased due to tumor purity?

To test whether our immune cell NES metric is biased due to tumor purity, we calculated tumor cell purity using the Sequenza⁹ tool based on our WES data. The estimated tumor cell purity was listed in **Supplementary Table 2**. The correlation between immune cell NES and tumor cell purity is now shown in **Response Figure 2**.

Overall, we observed no significant correlation between these two indexes, supporting an unbiased evaluation of immune cell NES in our analysis.

Response Figure 2. Scatterplot evaluating the correlation between immune-cell NES and the purity of tumor tissues. Tumor cell purity in our samples was determined using the Sequenza tool. Slope, Pearson fitting line; shaded area, 95% confident interval.

Figure 1:

What are the shaded area and the line in Figures 1E-F?

The shaded areas represent the 95% CI regions; we now include this information in the figure legend.

The “Machine Learning” panel of Figure 1A seems to have been copied and pasted from Hackl et al. (Nature Reviews Genetics, 2016), with obvious copyright issues.

Replace in Figure 1A “foci” with “loci”.

We apologize for this mistake. We have updated the “Machine Learning” panel with our own illustration and now cite Hackl et al.

Figure 1C: predicted neoantigens should be filtered for expressed ones (using RNA-seq data). A figure with the neoantigen burden (as 1E and 1F) would be helpful.

Thank you for your suggestion. Since only 44 of the 57 tissues were tested for RNA-Sequencing, we cannot annotate directly in Figure 1C to show which predicted neoantigen accompanied by detectable transcript. However, we provide this information in **Supplementary Table 8** to list all the predicted neoantigens with detectable transcript in RNA-Sequencing data. Neoantigen burdens across tumor loci are now presented as **Supplementary Figure 3**.

Figure 2:

Instead of dichotomizing the cytolytic activity into two classes, it would be better to plot it as continuous variable, like for gene expression levels.

We thank Reviewer #2 for this excellent suggestion. In our revised Figure 2A, we now code the cytolytic score of each individual tumor locus using a continuous color gradient.

Figure 3:

Figure 3A which variable was plotted (NES?).. The legend should also explain what are the shaded area and the line of Figure 3C.

Figure 3A plots the normalized ssGSEA score. We have added the label to the left of the revised figure. In Figure 3C, the shaded area represents 95% confident interval.

Response to Reviewer #3

The authors analyzed different regions of lung cancer for somatic mutations, TCR-beta repertoire and immune signature by DNA-seq and RNA-seq. By standard techniques (NetMHC), they predicted neoepitopes and tried to correlate neoepitopes with T cell clonal expansion and a signature of immune cytotoxicity. They found significant heterogeneity in different tumor regions, T cell expansion was associated with mutation burden but not with an immune cytotoxicity signature. These data contrast some prominently published papers, in which the immune cytotoxicity signature was correlated with decreased predicted neoepitopes. These earlier studies were performed with whole tumor tissue and, therefore, the current manuscript is important, since it demonstrates that tumor heterogeneity, at least of tumors with high mutational burden, introduces a complexity, which when ignored can lead to substantial false-positive results.

We appreciate Reviewer #3's comments on the value and potential clinical impact of our results.

A general critique on the current and all related studies, of which almost weekly appear new ones, is neoepitope prediction. None of these studies in human cancer demonstrated that a predicted neoepitope is indeed a neoepitope. Current algorithms predict the peptide-MHC binding affinity (but unlikely for all MHC I alleles) but cannot accurately predict proteasomal processing and not at all post-proteasomal trimming and whether sufficient peptide-MHC complexes reach the cell surface. With vaccinia virus as model, it was shown that only a minor fraction of HLA-A2 predicted epitopes were actually immunogenic in HLA-A2 transgenic mice (Assarsson et al. *J Immunol* 2007, 178:7890–7901). No reason to assume that it is different for cancer neoantigens. Thus, it is unclear how large the number of wrongly predicted neoepitopes is. A cut-off of IC50 of 500 nM is quite low, unclear whether such epitopes can elicit a T cell response in the cancer context. How do the results look like, if cut-off is 50 or 5 nM? That the T cells in the lung tumors are neoepitope-specific is possible but speculative and relies on data from few melanoma patients treated successfully with TILs.

Reviewer #3 elegantly outlined the core deficiency of all existing neoantigen prediction methods: while current algorithms can predict MHC-I binding affinities with very reasonable accuracy, MHC-I binding is not equivalent to antigenicity. Therefore, we agree that, like all other neoantigen prediction studies, our current analysis is far from ideal. The predicted affinity cutoff to qualify a neoantigen was set

at 500 nM (**Figure 1C** and **Supplementary Figure 2B**) in order to cover most possible neoantigens. As suggested by Reviewer #3, we have now performed this analysis using a more stringent cutoff of 50 nM predicted affinity (**Response Figure 3**, also **Supplementary Figure 3**). As shown below, these new results are almost identical to those obtained using the 500 nM threshold.

Response Figure 3 (also Supplementary Figure 3) Correlations between T cell repertoire or inflamed immune signature and number of neoantigens predicted using a more stringent parameter. Scatterplot showing correlation between the number of predicted neoantigens with binding strength < 50 nM, and expanded properties of the T cell repertoire. Shannon entropy index of T cell clonality (**A**) and Simpson diversity index (**B**) were used to depict the T cell repertoire composition. Enrichment of highly expanded clones results in higher values for clonality and Simpson diversity. (**C-E**) Correlation between predicted neoantigens (with binding strength < 50 nM) with expression of interferon-gamma, granzyme-A, and cytolytic activity (measured as the geometric mean of granzyme-A with perforin-1) in log2 of transcript per kilobase million (TPM). R, coefficient of Pearson correlation. Shaded areas represent the 95% confidence interval of fitting.

The terms tumor immunity, immunogenic or antigenic are poorly speculative. Immunity means “protection” but the T cells do not prevent tumor progression. “Reactivity” seems a more appropriate term (Nat Rev Cancer 12: 307-313, 2012). The authors have no indication that the expanded T cells ever were functional.

We appreciate Reviewer #3's precision in language. We agree that "reactivity" is a better word to describe this T cell behavior and have adopted this change throughout the revised manuscript.

It is not possible for this reviewer to go in detail through this vast amount of omics data and judge the bioinformatics. However, all conclusions of the current manuscript depend not only on accurate prediction of neoepitopes (see above) but also on accurate determination of the HLA class I haplotype. It seems that Optitype is not 100% accurate (Journal of Human Genetics (2017) 62, 397–405).

Indeed, in the original manuscript HLA typing was performed using Optitype, which has a reported prediction accuracy of 97%²⁸. Since accurate HLA haplotype is essential for neoantigen prediction, to validate our results we have re-analyzed our whole-exome data using HLA-VBseq²⁹. This algorithm produced results that are identical to those generated by Optitype. We have noted this additional analysis in the revised Material and Methods section.

For better judging the data, a Table with all mutations, predicted IC50 and RNAseq reads would be helpful.

We thank Reviewer #3 for this suggestion. As requested, we now include this information in **Supplementary Table 8** (predicted neoantigens with binding strengths) and **Supplementary Table 10** (RNA expression in transcripts per million).

Fig 1B-D are not understandable. Figure labels are too small.

We thank Reviewer #3 for pointing this out and have enlarged these figures and their associated labels. Also, more detailed descriptions have been added for Fig. 1B-D and provided in **Supplementary Figure 2**.

Lines 123-138: Lavin et al., 2017, Cell 169, 750–765 might be relevant to be cited.

Suppl Fig S3 should be labeled.

Reading the manuscript was difficult, since some Suppl Tables were wrongly labeled, e.g. Suppl Table 5 was missing.

We apologize for these omissions and mistakes and have corrected them in the revised manuscript as suggested.

In conclusion, bearing the above critiques in mind, the manuscript is interesting, because it puts into question current popular views and illustrates the problem with this type of analysis.

We again thank Reviewer #3 for her/his close and expert reading of our study.

REFERENCES

1. Balachandran, V.P., *et al.* Identification of unique neoantigen qualities in long-term survivors of pancreatic cancer. *Nature* **551**, 512-516 (2017).
2. Luksza, M., *et al.* A neoantigen fitness model predicts tumour response to checkpoint blockade immunotherapy. *Nature* **551**, 517-520 (2017).
3. Rizvi, N.A., *et al.* Cancer immunology. Mutational landscape determines sensitivity to PD-1 blockade in non-small cell lung cancer. *Science*. **348**, 124-128. doi: 110.1126/science.aaa1348. Epub 2015 Mar 1112. (2015).
4. Snyder, A., *et al.* Genetic basis for clinical response to CTLA-4 blockade in melanoma. *N Engl J Med* **371**, 2189-2199 (2014).
5. Van Allen, E.M., *et al.* Genomic correlates of response to CTLA-4 blockade in metastatic melanoma. *Science* **350**, 207-211 (2015).
6. Spranger, S., *et al.* Density of immunogenic antigens does not explain the presence or absence of the T-cell-inflamed tumor microenvironment in melanoma. *Proc Natl Acad Sci U S A* **113**, E7759-E7768 (2016).
7. Charoentong, P., *et al.* Pan-cancer Immunogenomic Analyses Reveal Genotype-Immunophenotype Relationships and Predictors of Response to Checkpoint Blockade. *Cell Rep.* **18**, 248-262. doi: 210.1016/j.celrep.2016.1012.1019. (2017).
8. Carbone, D.P., *et al.* First-Line Nivolumab in Stage IV or Recurrent Non-Small-Cell Lung Cancer. *N Engl J Med.* **376**, 2415-2426. doi: 2410.1056/NEJMoa1613493. (2017).

9. Favero, F., *et al.* Sequenza: allele-specific copy number and mutation profiles from tumor sequencing data. *Ann Oncol.* **26**, 64-70. doi: 10.1093/annonc/mdu1479. Epub 2014 Oct 1015. (2015).
10. Hugo, W., *et al.* Genomic and Transcriptomic Features of Response to Anti-PD-1 Therapy in Metastatic Melanoma. *Cell* **165**, 35-44 (2016).
11. Jung, S.H., *et al.* Genetic Progression of High Grade Prostatic Intraepithelial Neoplasia to Prostate Cancer. *Eur Urol* **69**, 823-830 (2016).
12. Eleveld, T.F., *et al.* Relapsed neuroblastomas show frequent RAS-MAPK pathway mutations. *Nat Genet* **47**, 864-871 (2015).
13. Blankenstein, T., Coulie, P.G., Gilboa, E. & Jaffee, E.M. The determinants of tumour immunogenicity. *Nat Rev Cancer* **12**, 307-313 (2012).
14. Rouillard, A.D., *et al.* The harmonizome: a collection of processed datasets gathered to serve and mine knowledge about genes and proteins. *Database (Oxford)*. **2016.**, baw100. doi: 110.1093/database/baw1100. Print 2016. (2016).
15. Davis, M.M., Tato, C.M. & Furman, D. Systems immunology: just getting started. *Nat Immunol* **18**, 725-732 (2017).
16. Barbie, D.A., *et al.* Systematic RNA interference reveals that oncogenic KRAS-driven cancers require TBK1. *Nature* **462**, 108-112 (2009).
17. Shen-Orr, S.S., *et al.* Cell type-specific gene expression differences in complex tissues. *Nat Methods* **7**, 287-289 (2010).
18. Zhao, Y. & Simon, R. Gene expression deconvolution in clinical samples. *Genome Med* **2**, 93 (2010).
19. Zhong, Y. & Liu, Z. Gene expression deconvolution in linear space. *Nat Methods* **9**, 8-9; author reply 9 (2011).
20. Shen-Orr, S.S. & Gaujoux, R. Computational deconvolution: extracting cell type-specific information from heterogeneous samples. *Curr Opin Immunol* **25**, 571-578 (2013).
21. Newman, A.M., *et al.* Robust enumeration of cell subsets from tissue expression profiles. *Nat Methods* **12**, 453-457 (2015).
22. Angelova, M., *et al.* Characterization of the immunophenotypes and antigenomes of colorectal cancers reveals distinct tumor escape mechanisms and novel targets for immunotherapy. *Genome Biol* **16**, 64 (2015).
23. Mandal, R., *et al.* The head and neck cancer immune landscape and its immunotherapeutic implications. *JCI Insight* **1**, e89829 (2016).
24. Riaz, N., *et al.* Tumor and Microenvironment Evolution during Immunotherapy with Nivolumab. *Cell* **171**, 934-949 e915 (2017).

25. McGranahan, N., *et al.* Clonal neoantigens elicit T cell immunoreactivity and sensitivity to immune checkpoint blockade. *Science*. **351**, 1463-1469. doi: 10.1126/science.aaf1490. Epub 2016 Mar 1463. (2016).
26. Blakely, C.M., *et al.* Evolution and clinical impact of co-occurring genetic alterations in advanced-stage EGFR-mutant lung cancers. *Nat Genet* **49**, 1693-1704 (2017).
27. Rooney, M.S., Shukla, S.A., Wu, C.J., Getz, G. & Hacohen, N. Molecular and genetic properties of tumors associated with local immune cytolytic activity. *Cell*. **160**, 48-61. doi: 10.1016/j.cell.2014.1012.1033. (2015).
28. Kiyotani, K., Mai, T.H. & Nakamura, Y. Comparison of exome-based HLA class I genotyping tools: identification of platform-specific genotyping errors. *J Hum Genet*. **62**, 397-405. doi: 10.1038/jhg.2016.1141. Epub 2016 Nov 1024. (2017).
29. Nariai, N., *et al.* HLA-VBSeq: accurate HLA typing at full resolution from whole-genome sequencing data. *BMC Genomics* **16**, S7. doi: 10.1186/1471-2164-11116-S1182-S1187. Epub 2015 Jan 1121. (2015).

Reviewers' Comments:

Reviewer #1:

Remarks to the Author:

The authors have not addressed my criticisms adequately.

comment 1. the logic of the data interpretation is still faulty. indeed, a number of biomarkers can contribute to identifying patients benefiting from immunotherapy but the authors seem to perseverate on "challenging" some perceived dogma of TMB when in fact no dogma exists. The authors need to make a better effort to adjust their interpretation, which is still not adequate.

2. validation sequencing was not done as I asked for

3. the machine learning concerns are not addressed. the approach is biased by preexisting selected input parameters. no rationale was given for inclusion other than ad hoc choice.

4. deconvolution does not perform well and in fact can perform quite poorly. the authors so not orthogonally validate with pathologic assessment.

Reviewer #2:

Remarks to the Author:

The authors satisfactorily addressed all issues raised in my previous review. Specifically, additional analyses, modifications of the manuscript and the figures contributed to clearly present the message and highlight novel findings.

Reviewer #3:

Remarks to the Author:

Except for one point, the authors have addressed all concerns: Supplementary Tables 6 and 8 are not readable and should be presented in an easy readable and informative form.

Sep 27, 2018

Response to Reviewer #1

The authors have not addressed my criticisms adequately

1. The logic of the data interpretation is still faulty. indeed, a number of biomarkers can contribute to identifying patients benefiting from immunotherapy but the authors seem to perseverate on "challenging" some perceived dogma of TMB when in fact no dogma exists. The authors need to make a better effort to adjust their interpretation, which is still not adequate.

We thank Reviewer #1 for urging us to clarify our message. As stated in our previous rebuttal letter, we do not deny the value of TMB or other biomarkers for predicting the efficacy of checkpoint blockade, even though the field generally agrees that the predictive value of all current biomarkers needs further improvement. We also agree with Reviewer #1 that, although TMB remains one of the most widely employed biomarkers, there is currently no prevailing dogma used to predict patient responsiveness to antitumor immunotherapy.

Rather than challenging the current clinical application of TMB, the central conclusion from our study is that intratumoral mutational and immune heterogeneity in NSCLC are much higher than has been previously appreciated. As a secondary consideration, our study presents direct evidence that TMB and its associated neoantigen load do not necessarily predict local anti-tumor immune responses in checkpoint blockade-naïve patients. This could be the result of many immunological mechanisms, one of which is the negative feedback elicited locally to suppress T cell responses, as presented in Fig. 3C.

In our previous revision, we also discussed two recent publications that closely align with our conclusions^{1,2}. First, in a study of pancreatic cancer immunotherapy-naïve patient cohorts, TMB failed to predict T cell immune reaction or prognosis¹. Moreover, when three clinical trials (two with anti-CTLA4 against melanoma^{3,4} and one with anti-PD1 against NSCLC⁵), were revisited, the general consensus was that neoantigen quality, which represents the antigenicity of mutations, has superior predictive value compared to neoantigen quantity (TMB)². It is worth noting that these three cohort analyses are the same ones used to establish the TMB-efficacy correlation; moreover, the same researchers who originally published the TMB-efficacy correlation are now making this adjustment. Their new antigen quality/fitness theory is reflected in our mutation evolution analysis (Fig. 3E), which shows that five of six patients with heterogeneous immune microenvironments display a dichotomy of mutations that separates cold versus hot regions into divergent evolutionary directions: other than homogenous (progenitor) mutations at every loci, hot and cold tumor regions did not share any common mutations. This suggests that the functional nature of a mutation-carrying protein may also play a role in determining the immunogenicity of neoantigens. Taken together, these results do not challenge the value of TMB as a biomarker for patient stratification. Rather, they provide a working hypothesis to explain why, as a single biomarker, TMB is imperfect.

Finally, we would again like to emphasize that our intention has never been to dispute the utility of TMB, which remains a valuable but limited biomarker for predicting patient responsiveness. Instead, our study emphasizes the remarkably high mutational

and immune heterogeneity present within primary untreated human lung tumors and its relevance toward guiding patient treatment decisions.

2. Validation sequencing was not done as I asked for

Repeating whole exome sequencing for the purpose of validation was not explicitly requested during the first round of review. Therefore, we provided rationale explaining why we trust the sequencing service provided by the clinical diagnostic company (Geneplus-Beijing).

NGS and its associated bioinformatics tools have significantly advanced in recent years, making it a stable and reproducible commodity for researchers. In some genetics studies, researchers return to first-generation technology (such as Sanger sequencing) to validate their final discoveries, typically being a very limited number of key pathogenic variants at the very end of the analysis pipeline. In oncology fields, especially in descriptive landscape-style research that looks into the overall profile of a tumor genome instead of a particular mutation, it is rarely seen necessary to perform orthogonal resequencing confirmation. In particular, both Foundation Medicine's FoundationOne CDx and MSKCC's MSK-IMPACT panels, which are both FDA-approved TMB-deriving NGS panels, do not include an orthogonal sequencing validation process. The sequencing platform used in our research, together with its mutation calling pipeline, is based on the same level of clinical environment built on top of commercial reagents and capture kits that have passed stringent quality control, follows industrial best-practice, and is thoroughly validated and approved by Chinese governmental agencies. It should therefore be trusted to produce the same level of accuracy and reproducibility.

Additionally, oncology studies are challenging and unique in the sense that clinical tumor samples are limited in quantity and virtually impossible to reproduce. In our study, each WES analysis consumes approximately 500 ng of DNA. As a result, orthogonal resequencing of each mutation at the genomic level for the purpose of technical confirmation is not practical.

Nevertheless, we understand Reviewer #1's concern and performed validation sequencing using an independent NGS platform. Accordingly, we picked five samples with sufficient remaining DNA, namely P002.T5, P002.T6, P004.T3, P019.T2, and P019.T3, and performed independent WES resequencing and data analysis with a second clinical diagnostic NGS service (GeneCast Biotechnology Co., Ltd.). This validation NGS employed a different WES capture kit and mutation calling was performed using a different bioinformatics pipeline. In addition, based on our original data, TMB from these five patient samples covered a wide mutational range.

Despite possible technical variation between the two facilities, the overall concordance rate across the two facilities is 94.34%, which is satisfactory. In addition, upon closer inspection of the discordant variants, we found that most are low frequency variants passing VAF threshold on one platform but not the other, which is reasonable. For individual samples, concordance across the two facilities can be harder to reach if the total number of variants is low. This is particularly true of sample P019.T3 where the concordance rate is only 57.14%. However, these discordances do not affect our overall conclusions since we focused on the aggregate numbers of mutations instead of individual ones: samples possessing high mutational loads have been confirmed to be indeed high and low mutational load samples are

indeed low (**Response Figure 1**). We have attached an Excel spreadsheet in this revision detailing this concordance analysis.

WES resequencing confirmation of five representative samples: summary								
Sample ID	Total Num of Mutations Detected by GenePlus	Mutations in GenePlus Human Exome v3 but not in GeneCast MedExome Panel	Remaining GenePlus Mutations in GeneCast Panel Region	Remaining GenePlus Mutations Not Detected by GeneCast	Mutations Detected by GenePlus and GeneCast	GeneCast Mutations Not Detected by GenePlus	Total Num of Mutations Detected by GeneCast	Overall Concordance
P002.T5	376	17	359	21	338	14	352	95.08%
P002.T6	380	16	364	20	334	13	347	95.29%
P004.T3	25	1	24	7	17	3	20	77.27%
P019.T2	19	0	19	1	18	3	21	90.00%
P019.T3	3	0	3	1	2	2	4	57.14%
Subtotal	803	34	769	50	709	35	744	94.34%

Response Figure 1. Validation sequencing of exome data by an independent lab. Scatterplot showing the number of somatic mutations as detected by two qualified sequencing labs (GeneCast and Geneplus). The regression line and correlation coefficient are labeled within the plot.

3. The machine learning concerns are not addressed. the approach is biased by preexisting selected input parameters. no rationale was given for inclusion other than ad hoc choice.

The purpose of machine learning is to reduce complexity and simplify categorization. We recognize that our immune gene panel is a subset of the whole transcriptome. It is true that our input variables are selected, which is also true of every other customized panel depicting the tumor microenvironment⁶⁻¹⁰. To our knowledge, in comparison to other recently published panels or algorithms⁶⁻¹⁰, our immune map is built with the highest number of variables. Our selection criteria is described in detail in the manuscript. Most importantly, the selective nature of immune panels, including ours and others', is based on previous findings and reasonable assumptions, which allows us to qualitatively describe relevant characteristics of the tumor microenvironment. A major goal of tumor immune microenvironment phenotyping is to guide candidate selection for immune-oncology treatment. In fact, many clinical trials have showed favorable outcomes for patients with "hot" tumor microenvironments, and each of these studies used a customized immune panel to define this "hotness"⁷⁻¹⁰.

4. Deconvolution does not perform well and in fact can perform quite poorly. the authors so not orthogonally validate with pathologic assessment.

Upon Reviewer #1's request, we re-quantified immune cell infiltration using the independent xCell method¹¹ as supporting evidence. The major results obtained using ssGSEA (Fig. 3) were validated by this new bioinformatics package (**Response Figure 2**).

Response Figure 2. Analysis of immune cell infiltration by the xCell method.

(A) xCell analysis identifying the relative infiltration of immune cell populations for 44 NSCLC tumor samples with available RNA-Sequencing data. The relative infiltration of each cell type is normalized into a z-score. (B) Correlation between infiltration of cell types executing anti-tumor immunity (ActCD4, ActCD8, TcmCD4, TcmCD8, TemCD4, TemCD8, Th1, Th17, ActDC, CD56^{bri}NK, NK, NKT) and cell types executing pro-tumor, immune suppressive functions (Treg, Th2, CD56^{dim}NK, imDC, TAM, MDSC, Neutrophil, and pDC). The shaded area represents the 95% confident interval. (C, D) Scatterplot showing correlation between heterogeneities of TMB and immune cell infiltration. X-axis: an average pairwise correlation coefficient was calculated to quantify the divergence of immune cell infiltration and presented as 1-value for clear visualization. Higher values, heterogeneous immune cell infiltration; lower values, homogeneous immune cell infiltration. Y-axis: variation in genomic mutations was determined by either coefficient of variation (C) or intra-tumoral heterogeneity (D). Higher values, heterogeneous TMB; lower values, homogeneous mutation pattern.

Response to Reviewer #2

The authors satisfactorily addressed all issues raised in my previous review. Specifically, additional analyses, modifications of the manuscript and the figures contributed to clearly present the message and highlight novel findings.

We thank Reviewer #2 for her/his evaluation and appreciation of our study.

Response to Reviewer #3

Except for one point, the authors have addressed all concerns: Supplementary Tables 6 and 8 are not readable and should be presented in an easy readable and informative form.

We thank Reviewer #3 for her/his comments regarding the novelty and clinical significance of our study. We have remade Supplementary Tables 6 and 8 to be more readable and browsable with Excel.

REFERENCES

1. Balachandran, V.P., *et al.* Identification of unique neoantigen qualities in long-term survivors of pancreatic cancer. *Nature* **551**, 512-516 (2017).
2. Luksza, M., *et al.* A neoantigen fitness model predicts tumour response to checkpoint blockade immunotherapy. *Nature* **551**, 517-520 (2017).
3. Snyder, A., *et al.* Genetic basis for clinical response to CTLA-4 blockade in melanoma. *N Engl J Med* **371**, 2189-2199 (2014).
4. Van Allen, E.M., *et al.* Genomic correlates of response to CTLA-4 blockade in metastatic melanoma. *Science* **350**, 207-211 (2015).
5. Rizvi, N.A., *et al.* Cancer immunology. Mutational landscape determines sensitivity to PD-1 blockade in non-small cell lung cancer. *Science*. **348**, 124-128. doi: 110.1126/science.aaa1348. Epub 2015 Mar 1112. (2015).
6. Charoentong, P., *et al.* Pan-cancer Immunogenomic Analyses Reveal Genotype-Immunophenotype Relationships and Predictors of Response to Checkpoint Blockade. *Cell Rep.* **18**, 248-262. doi: 210.1016/j.celrep.2016.1012.1019. (2017).
7. Sharma, P., *et al.* Nivolumab in metastatic urothelial carcinoma after platinum therapy (CheckMate 275): a multicentre, single-arm, phase 2 trial. *Lancet Oncol.* **18**, 312-322. doi: 310.1016/S1470-2045(1017)30065-30067. Epub 32017 Jan 30026. (2017).
8. Fehrenbacher, L., *et al.* Atezolizumab versus docetaxel for patients with previously treated non-small-cell lung cancer (POPLAR): a multicentre, open-label, phase 2 randomised controlled trial. *Lancet.* **387**, 1837-1846. doi: 1810.1016/S0140-6736(1816)00587-00580. Epub 02016 Mar 00510. (2016).
9. Muro, K., *et al.* Pembrolizumab for patients with PD-L1-positive advanced gastric cancer (KEYNOTE-012): a multicentre, open-label, phase 1b trial. *Lancet Oncol.* **17**, 717-726. doi: 710.1016/S1470-2045(1016)00175-00173. Epub 02016 May 00173. (2016).
10. Doi, T., *et al.* Safety and Antitumor Activity of the Anti-Programmed Death-1 Antibody Pembrolizumab in Patients With Advanced Esophageal Carcinoma. *J Clin Oncol* **8**(2017).

11. Aran, D., Hu, Z. & Butte, A.J. xCell: digitally portraying the tissue cellular heterogeneity landscape. *Genome Biol.* **18**, 220. doi: 210.1186/s13059-13017-11349-13051. (2017).

Reviewers' Comments:

Reviewer #1:

Remarks to the Author:

The authors have done a good job of answering my questions.